# An Hfq-dependent post-transcriptional mechanism fine tunes RecB expression in *Escherichia coli*

Irina Kalita[1,2,3]*, Ira Alexandra Iosub[1,4,5], Lorna McLaren[1,2], Louise Goossens[1,2], Sander Granneman[1,4], Meriem El Karoui[1,2]*

[1]Centre for Engineering Biology, University of Edinburgh, Edinburgh, United Kingdom; [2]Institute of Cell Biology, University of Edinburgh, Edinburgh, United Kingdom; [3]Max Planck Institute for Terrestrial Microbiology & Center for Synthetic Microbiology (SYNMIKRO), Marburg, Germany; [4]Institute of Quantitative Biology, Biochemistry and Biotechnology, University of Edinburgh, Edinburgh, United Kingdom; [5]The Francis Crick Institute, London, United Kingdom

## eLife Assessment

Combining experimental and computation approaches, this manuscript provides **convincing** evidence for a post-transcriptional mechanism that provides robust control over the protein expression level of RecB in *E. coli*. In addition to uncovering how DNA damage drives higher levels of RecB protein, this work also reveals **important** tenets for how broader mechanisms that suppress noise and underlie responsive tuning of protein levels can be achieved.

*For correspondence:
irykalita@gmail.com (IK);
Meriem.Elkaroui@ed.ac.uk (MEK)

Competing interest: The authors declare that no competing interests exist.

**Abstract** All living organisms have developed strategies to respond to chromosomal damage and preserve genome integrity. One such response is the repair of DNA double-strand breaks (DSBs), one of the most toxic forms of DNA lesions. In *Escherichia coli*, DSBs are repaired via RecBCD-dependent homologous recombination. RecBCD is essential for accurate chromosome maintenance, but its over-expression can lead to reduced DNA repair ability. This apparent paradox suggests that RecBCD copy numbers may need to be tightly controlled within an optimal range. Using single-molecule fluorescence microscopy, we have established that RecB is present in very low abundance at mRNA and protein levels. RecB transcription shows high fluctuations, yet cell-to-cell protein variability remains remarkably low. Here, we show that the post-transcriptional regulator Hfq binds to *recB* mRNA and downregulates RecB protein translation in vivo. Furthermore, specific disruption of the Hfq-binding site leads to more efficient translation of *recB* mRNAs. In addition, we observe a less effective reduction of RecB protein fluctuations in the absence of Hfq. This fine-tuning Hfq-mediated mechanism might have the underlying physiological function of maintaining RecB protein levels within an optimal range.

## Introduction

All living organisms rely on multiple molecular mechanisms to repair their chromosomes in order to preserve genome integrity. DNA double-strand breaks (DSBs), simultaneous lesions on both strands of the chromosome, are one of the most detrimental types of DNA damage because they can lead to loss of genetic information. If not repaired or repaired incorrectly, DSBs can trigger cell death, deleterious mutations and genomic rearrangements (*Wyman and Kanaar, 2006*). In addition to various exogenous sources such as ionizing radiation, mechanical stresses and DNA-damaging agents, DSBs

can occur during the normal cell cycle as a result of endogenous metabolic reactions and replication events (*Mehta and Haber, 2014*). For instance, replication forks often encounter obstacles leading to fork reversal, accumulation of gaps that are repaired by the RecFOR pathway (*Cox et al., 2000*) or breakage which has been shown to result in spontaneous DSBs in 18% of wild-type *Escherichia coli* cells in each generation (*Sinha et al., 2018*), underscoring the crucial need to repair these breaks to ensure faithful DNA replication.

In *E. coli*, DSBs are repaired by homologous recombination using an intact copy of the chromosome as a template. Initially, the multi-subunit nuclease/helicase enzyme RecBCD detects a double-strand DNA (dsDNA) end and degrades both strands. Upon recognition of a short DNA sequence, known as a *chi* ($\chi$) site (5'-GCTGGTGG-3'), the RecBCD complex stops cleaving the 3'-ended strand and initiates loading of the RecA protein onto the 3'-single-strand (ssDNA) overhang (*Spies et al., 2007*; *Wiktor et al., 2018*). This leads to the formation of a RecA-ssDNA nucleoprotein filament (*Arnold and Kowalczykowski, 2000*), which catalyses homology search and strand exchange (*Michel and Leach, 2012*).

Recent in vitro and in vivo experiments have highlighted the extraordinary processive DNA degradation activity of RecBCD upon recognition of a dsDNA end (*Roman et al., 1992*; *Bianco et al., 2001*; *Liu et al., 2013*). Indeed, RecBCD has been shown to process the chromosome for up to ~100 kb at ~1.6 kb/s in live bacteria (*Wiktor et al., 2018*). Such a potent DNA degradation activity is controlled by *chi* recognition (*Taylor et al., 2014*). However, because this recognition is probabilistic (*Cockram et al., 2015*), RecBCD can degrade large chromosome fragments before DNA repair is initiated. As a result, overproduction of RecBCD has been shown to sensitize wild-type cells to ultraviolet and gamma irradiations (*Dermić et al., 2005*). This strongly suggests that an excess of RecBCD imposes a threat to cells upon DNA damage. In contrast, RecBCD expression is essential upon DSB induction (*Emmerson, 1968*; *White et al., 2018*), hence too low levels might impair DNA repair. Thus, the observation that RecBCD can positively and negatively impact cell fitness upon DNA damage suggests that its activity needs to be tightly controlled.

While the biochemical and in vivo activities of RecBCD have been extensively studied (*Taylor et al., 2014*; *Amundsen et al., 2020*; *Cheng et al., 2020*; *Amundsen and Smith, 2019*; *Zananiri et al., 2019*; *Michel and Leach, 2012*; *Dillingham and Kowalczykowski, 2008* for review), less is known about the regulation of its expression. RecBCD has been reported to be expressed at very low levels (*Hickson et al., 1985*; *Taylor and Smith, 1980*) and is present in less than 10 molecules per cell (*Lepore et al., 2019*). In *E. coli*, the RecBCD subunits are encoded in the same locus by two operons. One operon controls the expression of RecC while the other is polycistronic and encodes PtrA, RecB, and RecD (*Hawley and McClure, 1983*; *Claverie-Martin et al., 1987*; *Figure 1A*). PtrA is a periplasmic protease with no known function in DNA repair, and the positioning of its gene likely results from a horizontal gene transfer event that has interrupted the usual order *recC–recB–recD* (*Cromie, 2009*). As sometimes found in bacterial operons (*Oppenheim and Yanofsky, 1980*; *Schümperli et al., 1982*), the ribosome-binding site (RBS) of *recB* is located within the coding sequence of *ptrA*, suggesting a potential translational coupling mechanism between these genes. Previous attempts to explore RecBCD expression upon DNA-damage conditions did not reveal any transcriptional regulation (*Hickson et al., 1985*; *Hickson et al., 1984*) and the *recBCD* genes were shown not to belong to the SOS-response regulon (*Wade et al., 2005*). Thus, RecBCD transcription is considered to be constitutive.

Given that RecBCD is essential for cell survival upon DSB formation and toxic when overproduced, it is likely that its expression needs to be tightly regulated to avoid fluctuations leading to cellular death or deleterious genomic rearrangements. However, its very low abundance, often leading to high stochastic fluctuations, could result in large cell-to-cell variability (*Paulsson, 2004*; *Uphoff et al., 2016*; *Elowitz et al., 2002*). Although these fluctuations might not play a significant role during normal cell growth, they may become crucial and determine cell fate upon stress conditions (*Raj and van Oudenaarden, 2008*; *Balaban et al., 2004*; *Harms et al., 2016*). Thus, the low abundance of RecBCD enzyme molecules raises the following questions: (1) to what extent do RecBCD numbers fluctuate, (2) are these fluctuations controlled, and (3) if this is the case, at which level (transcription, translation, mRNA, or protein degradation) is the control operating.

The lack of evidence of RecBCD transcriptional regulation does not preclude that other mechanisms may be involved in regulating its expression. For example, RNA-binding proteins (RBPs) have

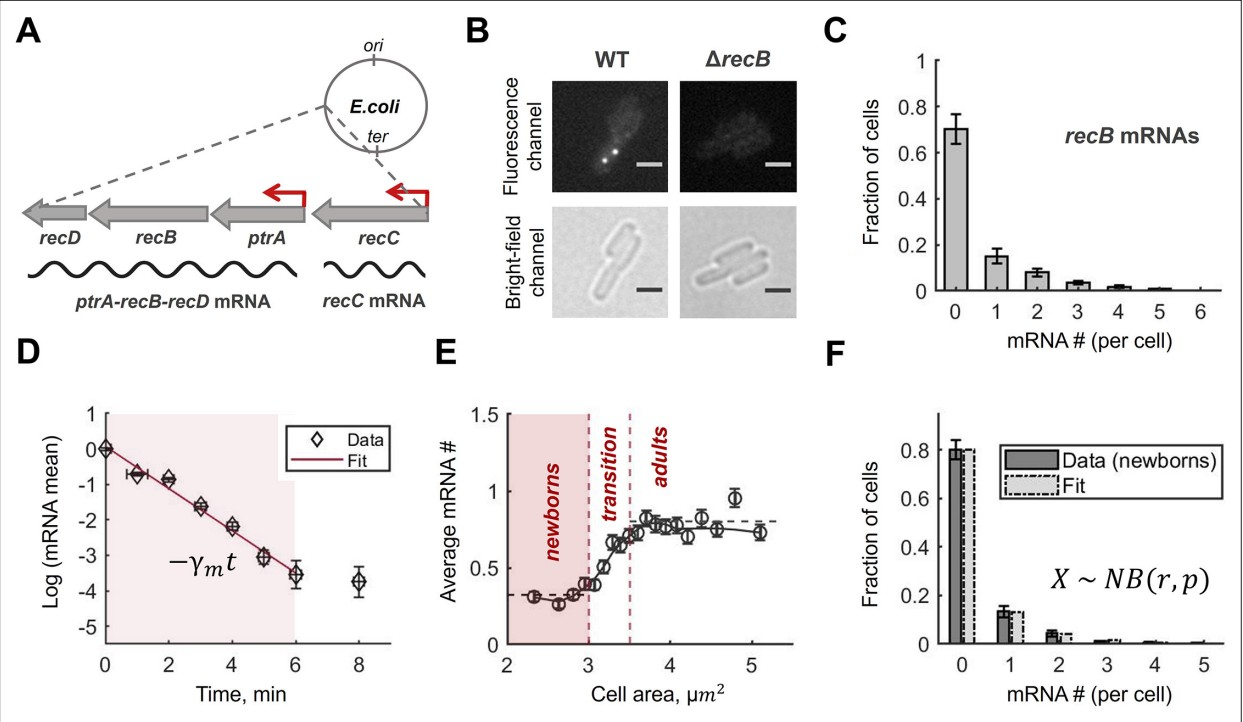

**Figure 1.** *recB* mRNAs are low abundant, short-lived and constitutively expressed. (**A**) Schematic description of the *recBCD* locus, its location on the *E. coli* chromosome and the corresponding mRNAs. (**B**) Examples of fluorescence and bright-field images of *recB* mRNA FISH experiments in wild-type (WT) and Δ*recB* strains. Scale bars represent 2 μm. (**C**) Total *recB* mRNA distribution quantified with single-molecule fluorescent in situ hybridization (smFISH) and presented in molecule numbers per cell. The histogram represents the average across three replicated experiments; error bars reflect the maximum difference between the repeats. Total number of cells, included in the distribution, is 15,638. (**D**) *recB* mRNA degradation rate measured in a time-course experiment where transcription initiation was inhibited with rifampicin. Mean mRNA counts, calculated from the total mRNA distributions for each time point, were normalized to the mean mRNA number at time *t = 0* and represented in the natural logarithmic scale. Vertical error bars represent the standard error of the mean (s.e.m.); horizontal errors are given by experimental uncertainty on time measurements. The shaded area shows the time interval used for fitting. The red line is the fitted linear function,$-\gamma_m t$, where $\gamma_m$ is the *recB* mRNA degradation rate. The final degradation rate was calculated as the average between two replicated time-course experiments (***Supplementary file 1G***). (**E**) *recB* mRNA molecule numbers per cell from the experiments in (**C**) shown as a function of cell area. The black circles represent the data binned by cell size and averaged in each group (mean ± s.e.m.). The solid line connects the averages across three neighbouring groups. Based on the mean mRNA numbers, all cells were separated into three sub-populations: *newborns*, *cells in transition*, and *adults*. (**F**) Experimental data from (**C**) conditioned on cell size (<3.0 μm², *newborns*; total cell number is 2180) fitted by a negative binomial distribution, *NB(r,p)*. The Kullback–Leibler divergence between experimental and fitted distributions is $D_{KL} = 0.002$.

The online version of this article includes the following figure supplement(s) for figure 1:

**Figure supplement 1.** Single-molecule fluorescent in situ hybridization (smFISH) image analysis performed by Spätzcell.

**Figure supplement 2.** Sensitivity of the single-molecule fluorescent in situ hybridization (smFISH) protocol allows for quantification of low-abundant *recB* mRNAs.

**Figure supplement 3.** Independent transcription from *recB* gene copies.

been shown to fine-tune gene expression both co- and post-transcriptionally during stress responses and adaptation to environmental changes (***Christopoulou and Granneman, 2022***; ***Kambara et al., 2018***). One of the most studied RBPs in *E. coli*, a Sm-like hexameric ring-shaped protein called Hfq, has been shown to regulate RNA stability and degradation, transcription elongation and translation initiation through various mechanisms (for review ***Nogueira and Springer, 2000***; ***Chao and Vogel, 2010***; ***Van Assche et al., 2015***; ***Holmqvist and Vogel, 2018***). In the most prevalent case, Hfq facilitates base-pairing between small non-coding RNAs (sRNAs) and target mRNAs which often results in mRNA translation and/or stability modulation (***Updegrove et al., 2016***). There is also increasing evidence that Hfq can control mRNA translation directly (***Kavita et al., 2018***; ***Chen and Gottesman, 2017***). This mechanism is typically based on altering the accessibility of the RBS of a target mRNA by direct binding near this region (***Sudo et al., 2022***; ***Sonnleitner and Bläsi, 2014***; ***Ellis et al., 2015***).

Here, we show that the expression of RecB is regulated post-transcriptionally in an Hfq-dependent manner in *E. coli*. By quantifying RecB mRNAs and proteins at single-molecule resolution, we found that constitutive transcription leads to low levels of *recB* mRNAs and a noisy distribution. In contrast, fluctuations in RecB protein levels are significantly lower than expected, given the very low number of proteins present in cells. We show that Hfq negatively regulates RecB translation and demonstrate the specificity of this post-transcriptional control in vivo. Furthermore, we provide evidence of the role of Hfq in reducing RecB protein fluctuations and speculate that this could contribute to a fine-tuning mechanism to control RecBCD molecules within an optimal range.

## Results

### *recB* mRNAs are present at very low levels and are short-lived

To quantitatively measure RecB expression, we first precisely quantified *recB* mRNAs using single-molecule fluorescent in situ hybridization (smFISH) (*Skinner et al., 2013*). We confirmed that our hybridization conditions are highly specific as *recB* mRNA quantification in a Δ*recB* strain resulted in a negligible error (~0.007 molecules per cell, *Figure 1B*, *Figure 1—figure supplement 1*). In wild-type cells, we detected a very low average level of *recB* transcripts: ~0.62 molecules per cell. Notably, ~70% of cells did not have any *recB* mRNAs (*Figure 1C*). To confirm that this result was not because of low sensitivity of the smFISH protocol, we over-expressed *recB* mRNAs from an arabinose-inducible low-copy plasmid and detected the expected gradual amplification of the fluorescence signal with increased arabinose concentration (*Figure 1—figure supplement 2*).

We next measured the degradation rate of *recB* transcripts in a time-series experiment. Initiation of transcription was inhibited with rifampicin and *recB* mRNA expression levels were quantified by smFISH at subsequent time points. By fitting the mean mRNA value evolution over time to an exponential function, we estimated the *recB* degradation rate as $\gamma_m$= 0.62 min$^{-1}$ ($CI_{95\%}$ [0.48, 0.75]) (*Figure 1D*). This corresponds to a *recB* mRNA lifetime of 1.6 min and is consistent with the genome-wide studies in *E. coli* where transcript lifetimes were shown to be on average ~2.5 min across all mRNA species (*Chen et al., 2015*; *Moffitt et al., 2016*).

### *recB* transcription is constitutive and weak

As noted previously, quantitative analysis of transcription at the single-cell level needs to take into account the variation of gene copy number during the cell cycle (*Wang et al., 2019*; *Peterson et al., 2015*; *Bryant et al., 2014*). Therefore, we analysed RNA molecule abundance using cell size as a proxy for the cell cycle. Specifically, we binned all bacteria in groups based on cell area and averaged mRNA numbers in each of those size-sorted groups (*Figure 1E*). A twofold increase in mRNA abundance was observed within a narrow cell-size interval (~3.0–3.5 μm$^2$), which is consistent with an expected doubling of the transcriptional output as a result of the replication of the *recB* gene. We further showed that *recB* transcription from sister chromosomes happens independently, in full agreement with the hypothesis of an unregulated gene (*Figure 1—figure supplement 3*). Taken together, this evidence indicates that *recB* mRNA levels follow a gene-dosage trend, strongly suggesting that *recB* is constitutively expressed (*Wang et al., 2019*).

Next, to infer *recB* transcription rate, we used a simple stochastic model of transcription where mRNAs are produced in bursts of an average size *b* at a constant rate $k_m$ and degraded exponentially at a rate $\gamma_m$ (Appendix 1). The expected probability distribution for mRNA numbers at steady state is given by a negative binomial (*Peccoud and Ycart, 1995*; *Raj et al., 2006*; *Shahrezaei and Swain, 2008*). We used maximum likelihood estimation to infer the parameters of the negative binomial distribution from the experimental data (*Figure 1F*, Materials and methods). For the inference, we restricted our analysis to newborn cells (*Figure 1E*) as this model does not take into account cell growth. We then inferred the burst size for the *recB* gene, *b* ~ 0.95, and the rate of transcription, $k_m$ = 0.21 min$^{-1}$ (*Supplementary file 1G*, Materials and methods). The latter is remarkably low and close to the lower bound reported range of transcription rates in *E. coli*, 0.17–1.7 min$^{-1}$ (*Proshkin et al., 2010*; *Bremer and Dennis, 2008*; *Vogel and Jensen, 1994*). Therefore, the low abundance of *recB* mRNAs can be explained by infrequent transcription combined with fast mRNA decay.

Variations in mRNA molecule numbers, as a result of their low abundance and short lifetimes, can significantly contribute to gene expression noise (*Paulsson, 2004*; *Sanchez and Golding, 2013*;

*Swain, 2004*). To evaluate fluctuations of *recB* transcription, we calculated a commonly used measure of noise, the squared coefficient of variation ($CV_m^2$), defined as the variance divided by the squared mean $\sigma_m$. The average $CV_m^2$ in newborn cells was found to be $CV_m^2 = 5.9$, almost twice larger than the lower bound given by a Poisson process ($CV^2_{Poisson} = 1/\langle m \rangle = 3.1$). This suggests that *recB* mRNA production is noisy and further supports the absence of a regulation mechanism at the level of mRNA synthesis.

## RecB proteins are present in low numbers and are long-lived

To determine whether the mRNA fluctuations we observed are transmitted to the protein level, we quantified RecB protein abundance with single-molecule accuracy in fixed individual cells using the Halo self-labelling tag (*Figure 2A, B*). The HaloTag has been translationally fused to RecB in a loop after Ser47 (*Lepore et al., 2019*) where it is unlikely to interfere with the formation of RecBCD complex (*Singleton et al., 2004*), the initiation of translation and conventional C-terminal-associated mechanisms of protein degradation (*Weber et al., 2020*). Consistent with minimal impact on RecB production and function, bacterial growth was not affected by replacing the native RecB with RecB-HaloTag, the fusion was fully functional upon DNA damage and no proteolytic processing of the construct was detected (*Lepore et al., 2019*). To ensure reliable quantification in bacteria with HaloTag labelling, the technique was previously verified with an independent imaging method and resulted in >80% labelling efficiency (*Lepore et al., 2019*; *Okumus et al., 2016*). In order to minimize the number of newly produced unlabelled RecB proteins, labelling and quick washing steps were followed by immediate chemical fixation of cells. By applying this approach, we obtained RecB protein distributions for cells of all sizes and confirmed its low abundance: 3.9 ± 0.6 molecules per cell (*Figure 2C*, *Figure 2—figure supplement 1*).

To test if RecB is actively degraded by proteases or mainly diluted as a result of cell growth and division, we estimated the RecB protein removal rate. This is usually obtained from either pulse-chase experiments using radio-labelled amino acids or detection of protein concentration decay after treatment with a translation inhibitor (*Eldeeb et al., 2019*; *Yewdell et al., 2011*; *Doherty et al., 2009*). While the first approach is not amenable to single-cell assays, the latter may affect the measurements on long timescales because of the physiological impacts of stopping all protein production. Therefore, we used an alternative approach (*Merrill et al., 2019*; *Yamaguchi, 2009*) where we pulse-labelled RecB molecules with the HaloTag ligand, then removed the excess of the dye by extensive washing and measured the abundance of the labelled molecules over time. Assuming protein removal can be described by a first-order reaction, we fitted the mean protein counts over time to an exponential function on a subset of time points (*Figure 2D*, shaded area). The decay rate was extracted from the fit and resulted in $\gamma_p = 0.015$ min$^{-1}$ ($CI_{95\%}$ [0.011, 0.019]). Comparison to the population growth rate in these conditions (0.017 min$^{-1}$) suggests that RecB protein is stable and effectively removed only as a result of dilution and molecule partitioning between daughter cells. This result is consistent with a recent high-throughput study on protein turnover rates in *E. coli*, where the lifetime of RecB proteins was shown to be set by the doubling time (*Gupta et al., 2024*).

## RecB protein cell-to-cell fluctuations are lower than expected

We then extended the transcription model described above by considering protein synthesis and removal. This is known as the *two-stage* or *mRNA–protein* model of gene expression (*Shahrezaei and Swain, 2008*; *Thattai and van Oudenaarden, 2001*; *Paulsson, 2005*). In this model, a protein is produced from an mRNA at a constant translation rate $k_p$ while protein dilution is implicitly modelled by a first-order chemical reaction with rate $\gamma_p$ (Appendix 1). As discussed earlier, we focused on newborn cells and estimated the rate of translation as 0.15 min$^{-1}$ ($CI_{95\%}$ [0.12, 0.18], *Supplementary file 1G*, Materials and methods). This value fell below the lower bound of the previously reported range of the translation rates in *E. coli* (0.25–1 min$^{-1}$) (*Proshkin et al., 2010*; *Bremer and Dennis, 2008*). Therefore, we concluded that *recB* mRNAs are translated at low levels compared to other proteins in the cell.

Next, we examined whether the two-stage model can reproduce the cell-to-cell variability in the protein levels observed in our experiments. To this end, we simulated single trajectories of protein abundance over time using the Stochastic Simulation Algorithm (SSA) (*Gillespie, 1977*) for the two-stage model with the estimated parameters (*Supplementary file 1G*). Strikingly, the simulated distribution had larger fluctuations than the experimentally measured one (*Figure 2E*). While both protein

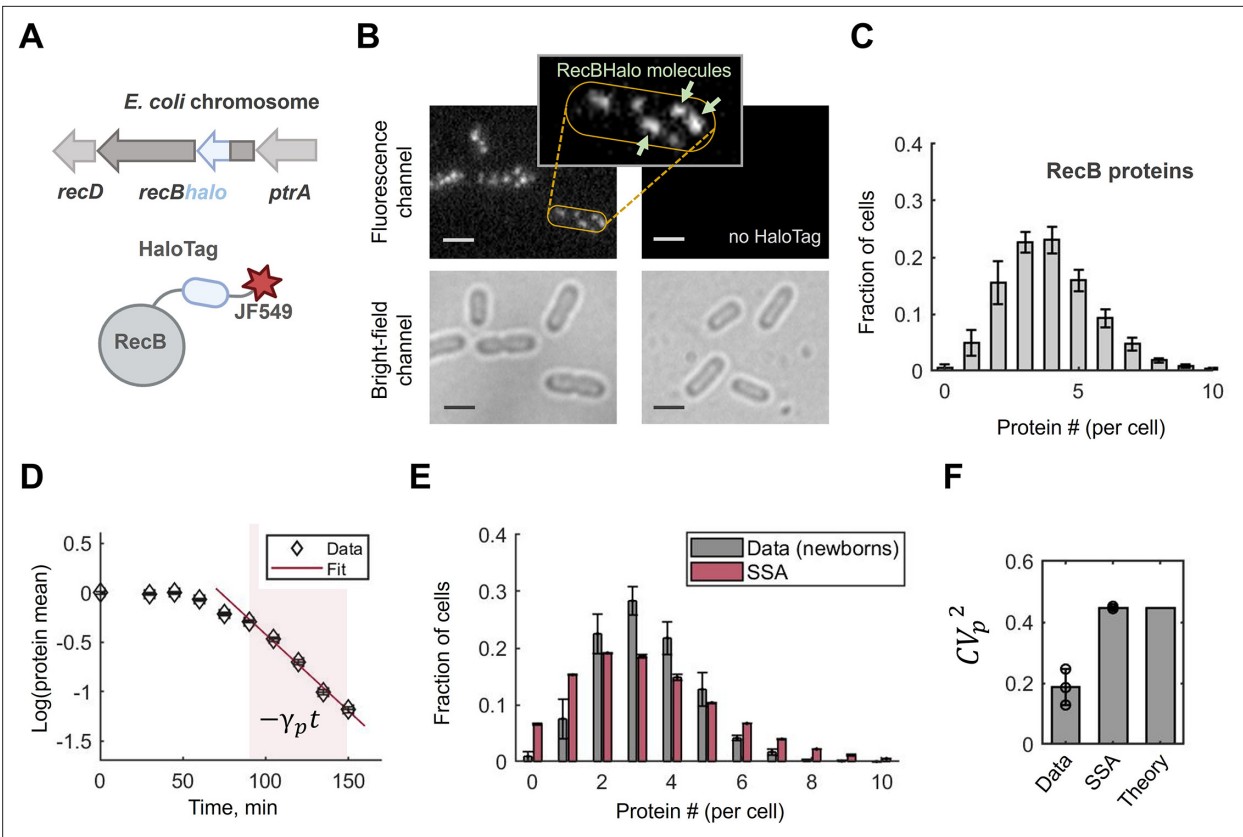

**Figure 2.** RecB proteins are low abundant, long-lived and show evidence of noise reduction. (**A**) Schematic of RecB-HaloTag fusion at the endogenous *E. coli* chromosomal locus of the *recB* gene. RecB-HaloTag is conjugated to the Janelia Fluor 549 dye (JF549). (**B**) Examples of fluorescence and bright-field Halo-labelling images for the strain with the RecBHalo fusion and its parental no HaloTag strain. Both samples were labelled with JF549 dye and the images are shown in the same intensity range. Scale bars represent 2 μm. Zoom-in: An example of a cell with five RecBHalo molecules (several single RecBHalo molecules are shown with light-green arrows). (**C**) Total RecB protein distribution quantified with Halo-labelling and presented in molecules per cell. The histogram represents the average of three replicated experiments; error bars reflect the maximum difference between the repeats. Total number of analysed cells is 10,964. Estimation of false positives in no HaloTag control resulted in ~0.3 molecules/cell. (**D**) RecB removal rate measured in a pulse-chase Halo-labelling experiment where the dye was removed at time *t = 0*. Mean protein counts, calculated from the total protein distributions for each time point, were normalized to the mean protein number at time *t = 0* and represented in the natural logarithmic scale. The shaded area shows the time interval used for fitting. The red line is the fitted linear function, $-\gamma_p t$, where $\gamma_p$ is the RecB removal rate. The final removal rate was calculated as the average between two replicated pulse-chase experiments *Supplementary file 1G*. (**E**) Comparison between the experimental RecB molecule distribution from (**C**) conditioned on cell size (in grey) and the results of Gillespie's simulations (SSA) for a two-stage model of RecB expression (in red). Parameters used in simulations are listed in *Supplementary file 1G*. The Kullback–Leibler divergence between the distributions is $D_{KL} = 0.15$. (**F**) Comparison of the coefficients of variation, $CV_p^2 = \sigma_p^2/\langle p \rangle^2$ for experimental (Data), simulated (SSA) data and analytical prediction (Theory). The experimental and simulated data from (**E**) were used; error bars represent standard deviations across three replicates; the black circles show the mean $CV_p^2$ in each experiment. Theoretical prediction was calculated using *Equation 1*.

The online version of this article includes the following figure supplement(s) for figure 2:

**Figure supplement 1.** Image analysis of Halo-labelling experiments performed with the modified version of Spätzcell.

means were equal because of the first-moment-based parameter estimation we used (Materials and methods), we found the standard deviations to be higher in simulation (2.2) than in experiments (1.5) (*Figure 2E*). In other words, a two-stage protein production model predicts a higher level of fluctuations than what is observed experimentally.

This phenomenon can also be analysed using the squared coefficient of variation of the protein distribution $CV_p^2$, which is defined as $\sigma_p^2/\langle p \rangle^2$. For the two-stage gene expression model with bursty transcription, it has been established that the following expression gives the coefficient of variation (Appendix 1) (*Shahrezaei and Swain, 2008*; *Thattai and van Oudenaarden, 2001*; *Paulsson, 2005*; *Pedraza and Paulsson, 2008*):

$$CV_p^2 = \frac{1}{\langle p \rangle} + \frac{1+b}{\langle m \rangle} \frac{\gamma_p}{\gamma_m + \gamma_p} \tag{1}$$

where $\langle p \rangle$ and $\langle m \rangle$ are the mean protein and mRNA counts (per cell), $b$ is the average (mRNA) burst size, $\gamma_m$ and $\gamma_p$ are the mRNA degradation and protein removal rates, respectively. We found that the experimental coefficient of variation ($CV^2_{Data} = 0.19 \pm 0.06$) is approximately half of the analytically predicted one ($CV^2_{Theory} = 0.45$) (*Figure 2F*). On the contrary, and as expected, both the theoretically predicted $CV^2_{Theory}$ and the one computed from the simulated data $CV^2_{SSA}$ are in good agreement. This deviation from the theoretical prediction implies a potential regulatory mechanism of RecB expression that actively suppresses protein production variation to maintain protein levels within a certain range.

## RecB translational efficiency is increased upon DNA damage

Next, we addressed whether there was any further evidence of this potential regulation once bacterial cells were exposed to stress. We hypothesized that RecB mRNA and/or protein levels may exhibit a concentration change upon DNA damage to stimulate DNA repair. To test this, we treated bacterial cells with a sub-lethal concentration of ciprofloxacin (4 ng/ml), an antibiotic that leads to DNA DSB formation (*Hooper and Jacoby, 2016*). After 2 hr of exposure to ciprofloxacin, we quantified proteins by Halo-labelling and mRNAs by smFISH (*Figure 3A*). We used mathematical modelling to verify that 2 hr of antibiotic exposure was sufficient to detect changes in mRNA and protein levels and for RecB mRNA and protein levels to reach a new steady state in the presence of DNA damage (Appendix 2). Because *E. coli* cells filament upon activation of the DNA damage response (*Figure 3B, C*), we measured single-cell distributions of mRNA and protein concentrations (calculated as molecule numbers normalized by cell area) rather than absolute numbers. Cell area is a reasonable proxy for cell volume as rod-shaped *E. coli* cells maintain their diameter while elongating in length upon ciprofloxacin treatment. Under DNA damage conditions, RecB protein concentration remained similar to the intact control (*Figure 3E*). In fact, the average cellular RecB concentration was surprisingly conserved: 2.28 mol/µm² ($CI_{95\%}$ [2.21, 2.32]) in perturbed condition and 2.12 mol/µm² ($CI_{95\%}$ [2.01, 2.32]) in intact cells. Additionally, we analysed RecB concentration in size-sorted groups of cells (*Figure 3—figure supplement 1b*). Regardless of moderate fluctuations in protein concentration over a cell cycle, both samples significantly overlap along the whole range of cell sizes.

In contrast, *recB* mRNA concentration was found to be decreased under exposure to ciprofloxacin (*Figure 3D*). Specifically, mRNA concentrations under DNA damage conditions are ~2.4-fold lower than in the unperturbed control (*Figure 3D*: inset): 0.12 mol/µm² ($CI_{95\%}$ [0.06, 0.17]) upon DNA damage versus 0.28 mol/µm² ($CI_{95\%}$ [0.23, 0.32]) in the control. We further confirmed the decrease in all cell-size-sorted groups (*Figure 3—figure supplement 1a*). As previously reported, the *recB* gene is not part of the SOS regulon (*Wade et al., 2005*), hence its transcription is not necessarily expected to be upregulated upon DNA damage. However, a lower level of *recB* mRNAs is an unforeseen observation and may be a result of lower concentration of *recB* gene copies and/or shortage of resources (such as RNA polymerases) upon SOS response activation.

We then computed the average number of proteins produced per mRNA in both conditions. The ratio between the average protein and mRNA concentrations in damaged cells is 19.7 ($CI_{95\%}$ [17.5, 24.2]) whereas it is 7.9 proteins/mRNA ($CI_{95\%}$ [7.5, 8.3]) in intact cells (*Figure 3F*). Thus, a ~2.5 increase in translational efficiency was detected under DSB damage conditions (*P*-value <0.001, two sample *t*-test). Taken together, our results show decoupling between mRNA and protein production upon DNA damage, suggesting the existence of a post-transcriptional regulatory mechanism that controls RecB protein level.

## Hfq controls the translation of *recB* mRNA

To identify the post-transcriptional mechanism controlling *recB* expression, we focused on Hfq because (1) it has been shown to regulate a vast number of mRNAs (*Tree et al., 2014*; *Zhang et al., 2013*; *Holmqvist et al., 2016*) and (2) its activity has been directly linked to the DNA damage and SOS responses (*Chen and Gottesman, 2017*; *Barreto et al., 2016*; *Pribis et al., 2019*).

To assess whether Hfq is involved in post-transcriptional regulation of RecBCD, we analysed the available transcriptome-wide dataset, wherein Hfq-binding sites were identified in *E. coli* using CLASH (cross-linking and analysis of cDNA) (*Iosub et al., 2020*). In a CLASH experiment, Hfq-RNA complexes are cross-linked in vivo and purified under highly stringent conditions, ensuring specific recovery of

Hfq RNA targets (*Iosub et al., 2020*). Analysis of the Hfq-binding profile across *recBCD* transcripts showed the interaction of Hfq with the *recBCD* mRNAs (*Figure 4A*). Hfq binding is mainly localized at two sites: one in close proximity to the RBS of the *ptrA* gene, and the second further downstream in the coding sequence of the *recB* gene. Interestingly, multiple trinucleotide Hfq-binding motifs, A-R(A/G)-N(any nucleotide) (*Tree et al., 2014*; *Link et al., 2009*) were found ~8 nucleotides upstream of the RBS of the *ptrA* sequence (*Figure 4—figure supplement 1*), suggesting that Hfq may control translation initiation and/or decay of the operon mRNA.

To test this hypothesis, we quantified RecB protein and mRNA levels in an *E. coli* strain lacking Hfq. Δ*hfq* mutant has been reported to have an increased average cell size and various growth defects (*Tsui et al., 1994*; *Muffler et al., 1997*; *Møller et al., 2002*). Indeed, we confirmed the increased cell size of our Δ*hfq* mutant compared to wild-type cells (*Figure 4—figure supplement 2a and b*). In our conditions, no difference between the growth rates of WT and the Δ*hfq* mutant was observed in the exponential phase of growth (*Figure 4—figure supplement 2e*).

Using the Halo-labelling technique, we performed RecB protein quantification in Δ*hfq*. Remarkably, we observed an increase of RecB protein concentration in the Δ*hfq* cells (*Figure 4B, C*). The Δ*hfq* mutant showed more fluorescent foci relative to the wild-type strain (*Figure 4B*). Even after normalization by cell area to take into account the larger cell size in Δ*hfq* cells, RecB concentration distribution was shifted towards higher values (*Figure 4C*). Quantification across five replicated experiments showed a statistically significant 30% increase in RecB protein concentration in the Δ*hfq* strain (*Figure 4C*: inset). The concentration was 2.67 mol/μm$^2$ ($CI_{95\%}$ [2.54, 2.80]) in Δ*hfq* compared to 2.17 mol/μm$^2$ ($CI_{95\%}$ [2.01, 2.32]) in the control.

We further quantified *recB* mRNA molecules in the Δ*hfq* strain using smFISH and analysed mRNA concentration distributions (*Figure 4D, E*). Although the average *recB* mRNA concentration is slightly decreased in the Δ*hfq*: 0.21 mol/μm$^2$ ($CI_{95\%}$ [0.08, 0.30]) versus 0.28 mol/μm$^2$ ($CI_{95\%}$ [0.23, 0.32]) in the wild-type cells, this difference is insignificant (*P*-value = 0.11, two sample *t*-test). These results were confirmed by RT-qPCR measurements, which did not show any significant changes for *recB* mRNA steady-state levels and lifetime (*Figure 4—figure supplement 2c and d*). Thus, we conclude that Hfq binding to the *recBCD* mRNA does not substantially alter the stability of the transcript.

The observed increase of RecB protein concentration without a significant change in *recB* mRNA steady-state levels is reflected in a rise in RecB translational efficiency (*Figure 4F*). The average level of proteins produced per mRNA in the Δ*hfq* mutant is 14.0 ($CI_{95\%}$ [13.0, 18.1]), which is two-fold higher than in the wild-type cells (*P*-value <0.05, two-sample *t*-test). Altogether, this demonstrates that Hfq alters RecB translation in vivo.

To test if Hfq regulation contributes to the noise suppression detected in the wild-type cells (*Figure 2F*), we looked at the level of fluctuations of RecB copy numbers in Δ*hfq*. To this end, we computed the difference between the analytically predicted noise $CV^2_{Theory}$ and the noise observed experimentally $CV^2_{Exp}$. Since in the Δ*hfq* mutant *recB* transcription is not affected, the theoretical noise was calculated using *Equation 1* assuming the same mRNA contribution (the second term of the equation). As seen in *Figure 4G*, the wild-type strain shows effective noise suppression as the theoretically predicted noise is larger than the one detected experimentally. Interestingly, this suppression is less effective in the Δ*hfq* mutant. In other words, in the absence of Hfq, the mismatch between the model and the experimental data is decreased in comparison to the native conditions of the wild-type cells (*P*-value = 0.0088, two sample *t*-test). This provides evidence that, in addition to the average RecB abundance, Hfq affects RecB protein fluctuations.

## RecB protein number distribution is recovered by Hfq complementation

To further confirm that Hfq regulates RecB expression, we complemented the Δ*hfq* mutant with a multi-copy plasmid that expresses Hfq, pQE-Hfq (*Morita and Aiba, 2019*). We confirmed the functionality of the Hfq protein expressed from the pQE-Hfq plasmid in our experimental conditions (*Figure 5—figure supplement 1d*). After complementation, the RecB protein concentration distribution was shifted towards lower values (*Figure 5B*) while no change of RecB protein level was detected in the Δ*hfq* cells carrying the backbone plasmid, pQE80L (*Figure 5—figure supplement 1a*). Although in our experimental conditions, Hfq is expressed at lower levels than in wild-type cells (transcription is ~ a third of the wild-type level, *Figure 5—figure supplement 1b*), RecB protein concentration was

nearly fully recovered to the wild-type conditions: 2.57 mol/μm² ($CI_{95\%}$ [2.50, 2.65]). We avoided full induction of Hfq from the plasmid as it can interfere with its self-regulation and lead to cell filamentation. We did not detect significant changes in the *recB* mRNA levels when expressing Hfq from the plasmid (*Figure 5—figure supplement 1c*), confirming that Hfq complementation impacts *recB* mRNA translation but not its stability.

## Sequestering Hfq leads to an increase in RecB production

To further ascertain the role of Hfq in controlling RecB expression, we characterized RecB expression while sequestering Hfq proteins away from its *recB* mRNA targets. Hfq is known to bind ~100 sRNAs (*Tree et al., 2014*; *Holmqvist et al., 2016*), some of which can reach high expression levels and/or have a higher affinity for Hfq and, thus, compete for Hfq binding (*Faigenbaum-Romm et al., 2020*; *Moon, 2011*). We reasoned that because *recB* mRNA is present in very low abundance, it could be outcompeted for Hfq binding by over-expression of a small RNA that efficiently binds Hfq, such as ChiX (*Moon, 2011*; *Małecka et al., 2015*). We constructed a multi-copy plasmid pZA21-ChiX to over-express ChiX and obtained a ~30-fold increase in ChiX expression relative to its native expression in wild-type cells (*Figure 5—figure supplement 2*). Given its already high abundance in wild-type cells,

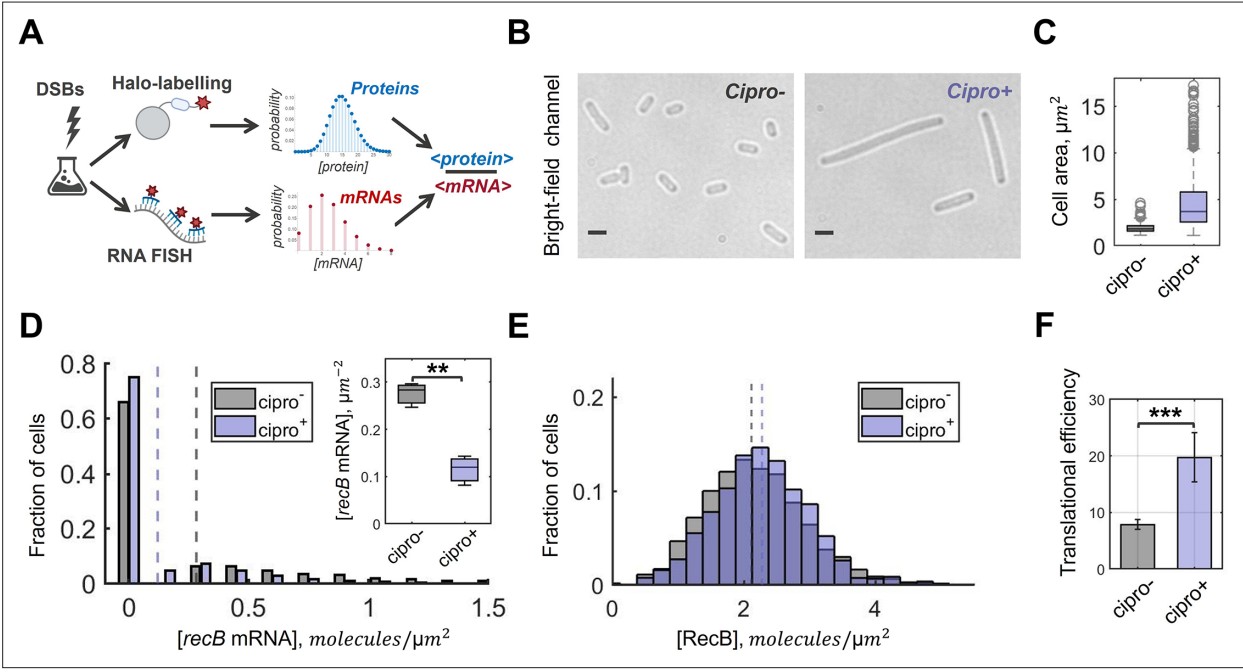

**Figure 3.** Translational efficiency of RecB is increased under DNA damage conditions. (**A**) Schematic of the experimental workflow. DSBs were induced with 4 ng/ml of ciprofloxacin for 2 hr, and protein (mRNA) quantification was performed with Halo-labelling (single-molecule fluorescent in situ hybridization, smFISH). Mean protein and mRNA concentrations were calculated from the distributions, and the average protein-to-mRNA ratio (translational efficiency) was estimated. (**B**) Examples of bright-field images for unperturbed (*cipro⁻*) and perturbed (*cipro⁺*) conditions. Scale bars represent 2 μm. (**C**) Box plot with cell area distributions for perturbed (blue) and unperturbed (grey) samples. The medians of cell area in each sample are 1.9 μm² (*cipro⁻*) and 3.6 μm² (*cipro⁺*). (**D**) *recB* mRNA concentration distributions quantified with smFISH in intact (grey) and damaged (blue) samples. The histograms represent the average of three replicated experiments. The medians of the *recB* mRNA concentrations are shown by dashed lines: 0.28 mol/μm² (*cipro⁻*) and 0.12 mol/μm² (*cipro⁺*). Total numbers of analysed cells are 11,700 (*cipro⁻*) and 6510 (*cipro⁺*). Inset: Box plot shows significant difference between the average *recB* mRNA concentrations (*P*-value = 0.0023, two-sample *t*-test). (**E**) RecB concentration distributions quantified with Halo-labelling in unperturbed and perturbed conditions. The histograms represent the average of three replicated experiments. The medians are shown by dashed lines: 2.12 mol/μm² (*cipro⁻*) and 2.28 mol/μm² (*cipro⁺*). Total number of analysed cells is 1534 (*cipro⁻*) and 683 (*cipro⁺*). The difference between the average RecB concentrations was insignificant (*P*-value = 0.36, two-sample *t*-test). (**F**) The average number of proteins produced per one mRNA in intact and damaged conditions. Average translational efficiency for each condition was calculated as the ratio between the mean protein concentration $c_p$ and the mean mRNA concentration $c_m$. The error bars indicate the standard deviation of the data; statistical significance between the conditions was calculated with two-sample *t*-test (*P*-value = 0.0001). In (**D, F**), significance marks stand for: $P \leq 0.05$ (*), $P \leq 0.01$ (**), $P \leq 0.001$ (***).

The online version of this article includes the following figure supplement(s) for figure 3:

**Figure supplement 1.** Analysis of *recB* mRNA and RecB protein concentrations and growth rate measurements under DNA damage.

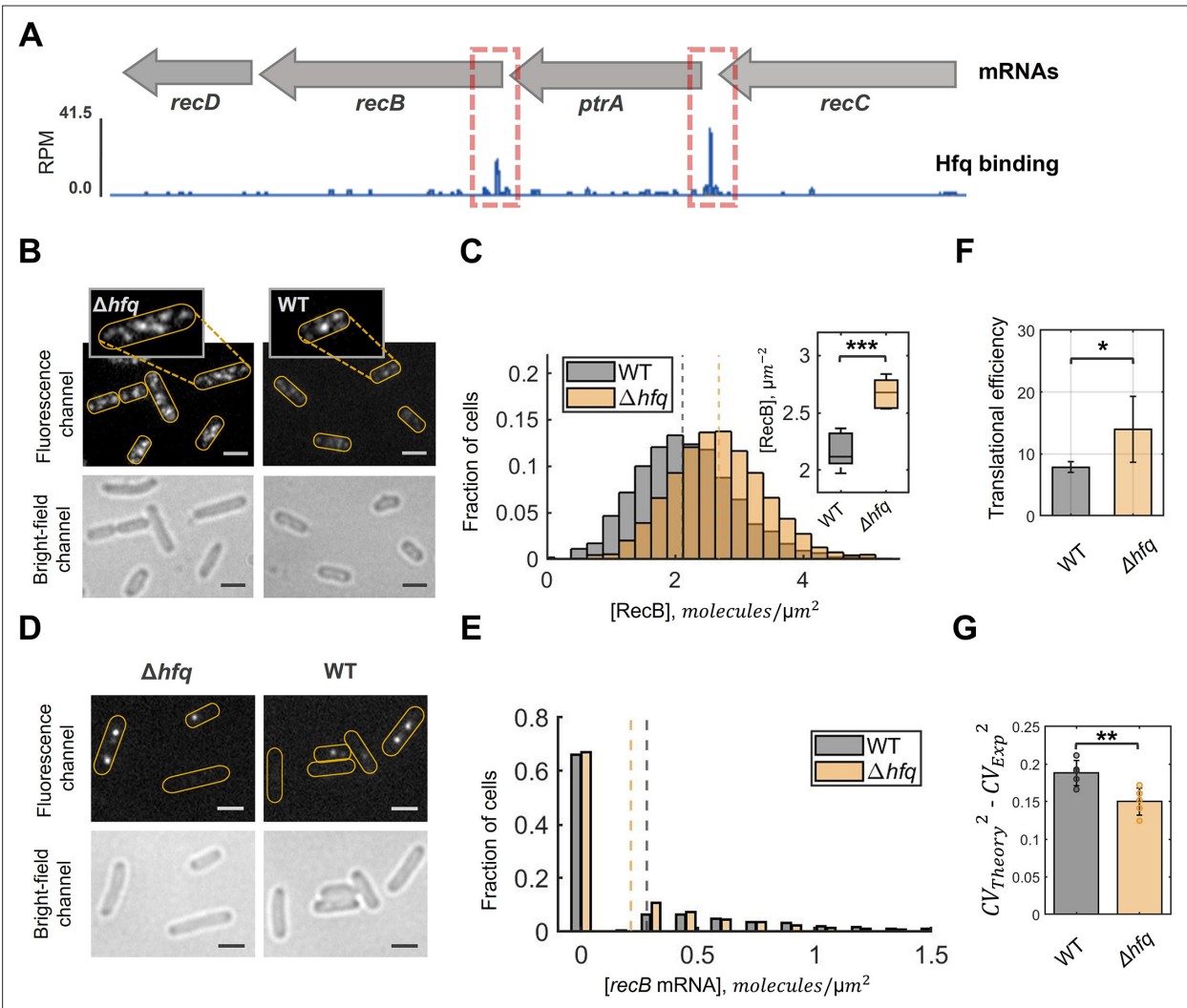

**Figure 4.** RecB expression is regulated by Hfq protein in vivo. (**A**) Genome browser track showing Hfq binding to *recC-ptrA-recB-recD* mRNAs. The coverage is normalized to reads per million (RPM). The major peaks of interest are highlighted by red dashed boxes. (**B**) Examples of fluorescence and bright-field images of RecB quantification experiments in Δ*hfq* and wild-type strains. Yellow outlines indicate rough positions of bacterial cells in the fluorescence channel. Scale bars represent 2 μm. Both fluorescence images are shown in the same intensity range while different background modulation was applied in the zoom-in figures (for better spots visualization). (**C**) RecB concentration distributions quantified with Halo-labelling in wild-type cells and the Δ*hfq* mutant. The histograms represent the average of five replicated experiments. The medians are shown by dashed lines: 2.12 mol/μm² for WT and 2.68 mol/μm² for Δ*hfq*. Total number of analysed cells is 3324 (WT) and 2185 (Δ*hfq*). Inset: Box plot shows significant difference (*P*-value = 0.0007) between the average RecB concentrations in wild-type and Δ*hfq* cells verified by two-sample *t*-test. (**D**) Examples of fluorescence and bright-field images of *recB* mRNA FISH experiments in Δ*hfq* and wild-type cells. Yellow outlines indicate rough positions of bacterial cells in the fluorescence channel. Scale bars represent 2 μm. (**E**) *recB* mRNA concentration distributions quantified with single-molecule fluorescent in situ hybridization (smFISH) in the Δ*hfq* mutant and wild-type cells. The histograms represent the average of three replicated experiments. The medians are shown by dashed lines: 0.28 mol/μm² for WT and 0.21 mol/μm² for Δ*hfq*. The total number of analysed cells is 7270 (WT) and 5486 (Δ*hfq*). The insignificant difference between the average *recB* mRNA concentrations in both strains was verified by two-sample *t*-test (*P*-value = 0.11). (**F**) RecB translational efficiency in wild-type and Δ*hfq* cells. Average translational efficiency (for each strain) was calculated as the ratio between the mean protein concentration $c_p$ and the mean mRNA concentration $c_m$ across replicated experiments. The error bars indicate the standard deviation of the data; statistical significance between the strains was calculated with two-sample *t*-test (*P*-value = 0.014). (**G**) The difference between theoretically predicted ($CV^2_{Theory}$) and experimentally measured ($CV^2_{Exp}$) coefficient of variation (squared). The theoretical values were computed according to 1. Error bars represent s.e.m. calculated from the replicated experiments. Statistical significance between the samples was calculated with a two-sample *t*-test (*P*-value = 0.0088). In (**C, F, G**), significance marks stand for: $P \leq 0.05$ (*), $P \leq 0.01$ (**), $P \leq 0.001$ (***).

The online version of this article includes the following figure supplement(s) for figure 4:

**Figure supplement 1.** The main Hfq-binding site within the *ptrA* gene.

**Figure supplement 2.** Cell size analysis, RT-qPCR quantification, and growth rate measurements in the Δ*hfq* mutant.

ChiX overproduction has been demonstrated to effectively sequester a large number of Hfq molecules (*Sudo et al., 2022*; *Ellis et al., 2015*; *Mandin and Gottesman, 2010*).

Wild-type cells carrying the pZA21-ChiX plasmid (or the backbone plasmid) were grown to mid-exponential phase, and RecB mRNA and protein levels were quantified with RT-qPCR and Halo-labelling, respectively. No significant changes were detected at the *recB* transcript levels when ChiX was overproduced (*Figure 5—figure supplement 2*). In contrast, a significant increase in RecB protein production was detected upon ChiX over-expression (*Figure 5C*). Indeed, RecB concentration in the cells where ChiX was over-expressed almost fully overlapped with the distributions in the Δ*hfq* mutant. This suggests that when fewer Hfq molecules are available for RecB regulation because ChiX is overproduced, RecB is translated at a higher rate. A similar experiment with a plasmid that carried a different sRNA, CyaR, which is not expected to titrate Hfq, did not change RecB protein concentration. This indicates that ChiX over-expression specifically impairs RecB post-transcriptional regulation.

To exclude the possibility of a direct ChiX involvement in RecB regulation, we characterized RecB expression in Δ*chiX*. No significant change relative to wild-type cells was detected in RecB mRNA or protein levels (*Figure 5—figure supplement 3*). Thus, we concluded that ChiX does not affect RecB translation. Instead, titration of Hfq proteins from a shared pool by ChiX sRNAs leads to the increased availability of *ptrA-recB* RBS for ribosomes and, thus, results in more efficient RecB translation.

## Deletion of an Hfq-binding site results in increased RecB translational efficiency

To further investigate the mechanism of RecB translation, we tested the specificity of Hfq interaction with the mRNA of *recB* without affecting other functions of Hfq. Based on the localization of the Hfq-binding uncovered in the CLASH data (*Figure 5A*), we deleted a 36 nucleotide region (5′-TTAA CGTGTTGAATCTGGACAGAAAATTAAGTTGAT-3′) located in 5′UTR of the operon, in front of *ptrA* gene. This region showed the highest enrichment of Hfq binding (*Figure 4—figure supplement 1*). It is closely located to the RBS site and 76 nucleotides downstream of the promoter sequence of *ptrA-recB*. To get a complete picture of RecB expression in the mutant strain carrying this deletion (which we further refer to as the *recB*-5′UTR* mutant), we performed single-cell quantification of both mRNA and protein levels of RecB and quantified its translational efficiency.

Firstly, we quantified *recB* mRNA molecules in the *recB*-5′UTR* strain using smFISH (*Figure 5D*). We detected a significant decrease in the concentration of mRNAs in the strain with the modified sequence: 0.14 mol/μm² ($CI_{95\%}$ [0.07, 0.17]) compared to 0.28 mol/μm² ($CI_{95\%}$ [0.23, 0.32]) in the wild type. This is not an unexpected outcome as the 5′-untranslated region of a transcript can control its degradation and stability (*Chen et al., 2022*; *Emory et al., 1992*). Despite the mRNA steady-state level decreasing two-fold in the modified *recB*-5′UTR* strain, only a slight decrease in RecB protein concentration was detected in comparison with the wild type (*Figure 5E*): 1.96 mol/μm² ($CI_{95\%}$ [1.56, 2.36]) in *recB*-5′UTR* and 2.17 mol/μm² ($CI_{95\%}$ [2.01, 2.32]) in the wild type.

We calculated the translational efficiency of *recB* transcripts based on our single-cell mRNA and protein quantification. The resulting translational efficiency in the *recB*-5′UTR* strain was found to be significantly higher than in the wild-type cells: 16.3 ($CI_{95\%}$ [14.3, 19.7]) in *recB*-5′UTR*, whereas it is 7.9 ($CI_{95\%}$ [7.5, 8.3]) proteins/mRNA in the wild type (*Figure 5F*). Remarkably, the RecB translational efficiency in the *recB*-5′UTR* strain is comparable with the values observed in the Δ*hfq* mutant (a two-sample *t*-test confirmed that the difference is not significant). Thus, when the main Hfq-binding site was deleted, the lower abundance of the mRNA was compensated by more efficient translation. This demonstrates that the translation of *recB* mRNA is limited by a specific interaction with Hfq.

## Discussion

Despite its essential role in DSB repair, over-expression of RecBCD decreases cell viability upon DNA damage induction (*Dermić et al., 2005*). Such a dual impact on cell fitness suggests the existence of an underlying regulation that controls RecBCD expression within an optimal range. Using state-of-the-art single-molecule mRNA and protein quantification, we (1) characterize RecB expression at transcriptional and translational levels in normal and DNA-damaging conditions and (2) describe a novel post-transcriptional mechanism of RecB expression mediated by the global regulator Hfq in vivo. Utilizing stochastic modelling, we provide evidence that Hfq contributes to suppressing fluctuations of

RecB molecules. To our knowledge, this is the first experimental evidence of an RNA-binding protein involved in noise suppression in bacterial cells under native conditions.

While a few recent studies have shown evidence for direct gene regulation by Hfq in a sRNA-independent manner (*Chen and Gottesman, 2017*; *Sudo et al., 2022*; *Sonnleitner and Bläsi, 2014*; *Ellis et al., 2015*; *Salvail et al., 2013*), we attempted to investigate whether a small RNA could be involved in the Hfq-mediated regulation of RecB expression. We tested Hfq mutants containing point mutations in the proximal and distal sides of the protein, which were shown to disrupt either binding with sRNAs or with ARN motifs of mRNA targets, respectively (*Zhang et al., 2013*; *Watkins and Arya,*

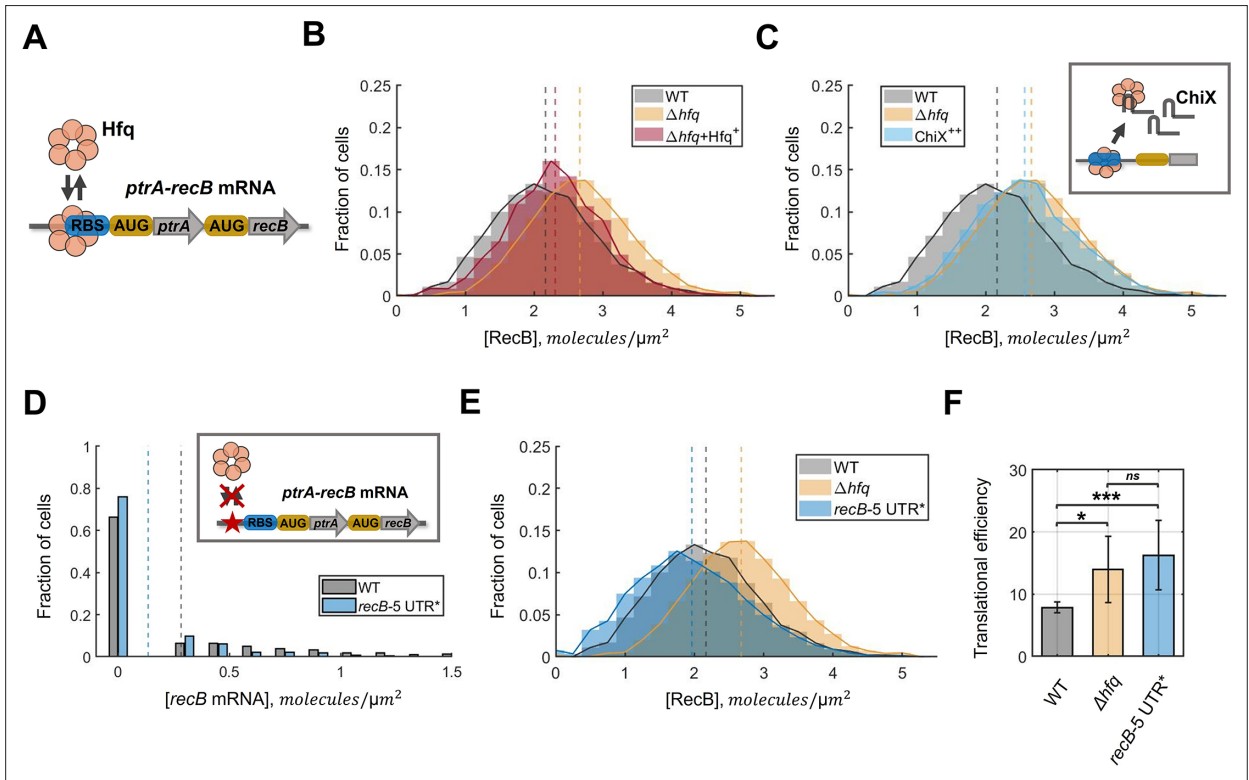

**Figure 5.** Specific alteration of RecB translation by Hfq in vivo. (**A**) A model of Hfq downregulating RecB translation by blocking the ribosome-binding site of the *ptrA-recB* mRNA. (**B**) Partial complementation of RecB expression to the wild-type level demonstrated by expression of a functional Hfq protein from a multicopy plasmid (pQE-Hfq) in Δ*hfq* (shown in red). The histogram represents the average of two replicated experiments. RecB concentration histograms in wild type and Δ*hfq* mutant from *Figure 4C* are shown for relative comparison. Dashed lines represent the average RecB concentration in each condition. (**C**) An increase in RecB protein production caused by sequestering of Hfq proteins with highly abundant small RNA ChiX (shown in light blue). The histogram represents the average of two replicated experiments. RecB concentration histograms in wild type and Δ*hfq* mutant from *Figure 4C* are shown for relative comparison. Dashed lines represent the average RecB concentration in each condition. (**D**) *recB* mRNA concentration distribution quantified with single-molecule fluorescent in situ hybridization (smFISH) in the strain with the deletion of the main Hfq-binding site *recB*-5'UTR* (shown in blue). The histogram represents the average of three replicated experiments. The *recB* mRNA distribution in the wild-type cells is shown for relative comparison. The medians are indicated by dashed lines. An approximate location of the removed sequence is schematically shown in the insert (red star). (**E**) RecB protein concentration distribution quantified with RecBHalo-labelling in *recB*-5'UTR* strain (shown in blue). The histogram represents the average of three replicated experiments. RecB concentration histograms in wild type and Δ*hfq* mutant from *Figure 4C* are shown for relative comparison. Dashed lines represent the average RecB concentration in each condition. (**F**) Translational efficiency of RecB in wild type, Δ*hfq* mutant and the strain with the deletion of the main Hfq-binding site, *recB*-5'UTR*. Average translational efficiency was calculated as the ratio between the mean protein concentration $c_p$ and the mean mRNA concentration $c_m$ across replicated experiments. The error bars indicate the standard deviation of the data; statistical significance between the samples was calculated with a two-sample *t*-test. The *P*-value for WT and *recB*-5'UTR* is 0.0007; while the difference between Δ*hfq* and *recB*-5'UTR* is non-significant (*P*-value >0.05). Significance marks stand for: $P \leq 0.05$ (*), $P \leq 0.01$ (**), $P \leq 0.001$ (***).

The online version of this article includes the following figure supplement(s) for figure 5:

**Figure supplement 1.** Controls for Hfq complementation experiment.

**Figure supplement 2.** Controls for ChiX over-expression experiment.

**Figure supplement 3.** RecB protein and mRNA quantification in Δ*chiX*.

*2023*). Hfq mutated in either proximal (K56A) or distal (Y25D) faces was expressed from a plasmid in Δ*hfq* background. In both cases, Hfq expression was confirmed with qPCR and did not affect *recB* mRNA levels (*Appendix 3—figure 1b*). When the proximal Hfq-binding side (K56A) was disrupted, RecB protein concentration was nearly similar to that obtained in Δ*hfq* (*Appendix 3—figure 1a*, top panel). This observation suggests that the repression of RecB translation requires the proximal side of Hfq, and that a small RNA is likely to be involved as small RNAs (Classes I and II) were shown to predominantly interact with the proximal face of Hfq (*Schu et al., 2015*). When we expressed Hfq mutated in the distal face (Y25D), which is deficient in binding to mRNAs, less efficient repression of RecB translation was detected (*Appendix 3—figure 1a*, bottom panel). This suggests that RecB mRNA interacts with Hfq at this position. We did not observe full de-repression to the Δ*hfq* level, which might be explained by residual capacity of Hfq to bind its *recB* mRNA target in the point mutant (Y25D) (either via the distal face with less affinity or via the lateral rim Hfq interface). Taken together, these results suggest that Hfq binds to *recB* mRNA and that a small RNA might contribute to the regulation. Therefore, we next attempted to look for potential candidates of sRNAs, using the available Hfq CLASH and RIL-seq datasets (*Iosub et al., 2020*; *Melamed et al., 2016*). CyaR and ChiX sRNAs were identified as potentially interacting with the *ptrA-recB-recD* mRNA. However, we did not find any significant changes in RecB expression when CyaR was over-expressed (*Figure 5—figure supplement 2*) and we showed that ChiX was involved indirectly by sequestering Hfq (*Figure 5*, *Figure 5—figure supplement 3*). Altogether, we could neither rule out the possibility of another small RNA candidate nor identify one involved, but the low abundance of the *recB* transcripts is likely to make it particularly difficult to detect RNA-RNA interactions in bulk experiments, making it necessary to develop a specific approach with single-molecule resolution.

Hfq-mediated control of RecB expression may be just a hint of a larger bacterial post-transcriptional stress–response programme to DNA damage. Indeed, post-transcriptional regulation has already been shown to impact DNA repair and genome maintenance pathways (*Barreto et al., 2016*; *Pribis et al., 2019*; *Mandin and Gottesman, 2009*; *Modi et al., 2011*). For instance, the DNA mismatch repair protein, MutS, was shown to be regulated by a sRNA-independent Hfq-binding mechanism and its translation repression was linked to increased mutagenesis in stationary phase (*Chen and Gottesman, 2017*). In addition, a modest decrease (~30%) in Hfq protein abundance has been seen in a proteomic study in *E. coli* upon DSB induction with ciprofloxacin (*Li et al., 2018*). While Hfq is a highly abundant protein, it has many mRNA and sRNA targets, some of which are also present in large amounts (*Zhang et al., 2003*). As recently shown, the competition among the targets over Hfq proteins results in unequal (across various targets) outcomes, where the targets with higher Hfq-binding affinity have an advantage over the ones with less efficient binding (*Faigenbaum-Romm et al., 2020*). In line with these findings, it is conceivable that even modest changes in Hfq availability could result in significant changes in gene expression, and this could explain the increased translational efficiency of RecB under DNA damage conditions. Therefore, Hfq-mediated regulation might play a larger role in DNA repair pathways than previously considered. As many metabolic pathways are regulated through Hfq (*Guisbert et al., 2007*; *Dos Santos et al., 2020*; *Večerek et al., 2003*), the control of RecB (and other DNA repair proteins) expression may even provide a way of coordinating DNA repair pathways with other cellular response processes upon stress.

Mechanisms similar to the one described in our study might underlie the expression networks of other DNA repair proteins, many of which are known to be present in small quantities. In this regard, it is worth emphasizing the valuable methodological insights which can be taken from our study. By utilizing single-molecule mRNA and protein quantification techniques, we show the importance of detailed investigation of both mRNA and protein levels when establishing novel regulation mechanisms of gene expression. Indeed, measuring the protein level only would not be sufficient to draw the correct interpretation, for example, in the case of *recB*-5'UTR* mutants (*Figure 5*). Moreover, single-molecule measurements combined with stochastic modelling allow access to cell-to-cell variability, which can be a powerful and, possibly, the only approach to investigate the regulation mechanisms of gene expression for low-abundant molecules.

In agreement with a previous report (*Dermić et al., 2005*), we confirm that RecBCD overproduction led to less efficient repair when DSBs were induced using ciprofloxacin (*Appendix 3—figure 2*). We note that another study did not detect any toxicity of RecBCD overproduction upon DNA damage (*Dykstra et al., 1984*). However, this is likely because the level of UV irradiation was too low to detect

a significant reduction in cell viability. We only observed the effect at a relatively high level of ciprofloxacin exposure (>10 ng/ml with the minimum inhibitory concentration at 16 ng/ml in our conditions). It is worth noting that the observed decrease in cell viability upon DNA damage was detected for relatively drastic perturbations such as *recB* deletion and RecBCD overexpression. Verifying these observations in the context of more subtle changes in RecB levels would be important for further investigation of the biological role of the uncovered regulation mechanism. However, the extremely low numbers of RecB proteins make altering its abundance in a refined, controlled, and homogeneous across cells manner extremely challenging and would require the development of novel synthetic biology tools.

In addition to the regulation of the average number of proteins per cell, fluctuation control might be crucial for a quasi-essential enzyme with very low abundance, such as RecBCD. Our analysis of RecB fluctuations suggests that Hfq is playing a role in protein copy number noise reduction (*Figure 4G*). Post-transcriptional regulation has been proposed theoretically as an effective strategy for reducing protein noise (*Swain, 2004*; *Shimoni et al., 2007*; *Singh, 2011*; *Bokes et al., 2021*; *Mehta et al., 2008*). Conceptually, this is achieved via buffering stochastic fluctuations of mRNA transcription and enabling more rapid response. Based on these principles, suppression of protein fluctuations via post-transcriptional control has been demonstrated in synthetically engineered systems in bacteria, yeast and mammalian cells (*Siciliano et al., 2013*; *Kelly et al., 2018*; *Mundt et al., 2018*). However, do cells employ these strategies under native conditions? Suppressed fluctuations of protein expression were experimentally detected for hundreds of microRNA-regulated genes in mammalian cells (*Schmiedel et al., 2015*). Remarkably, those genes were expressed at a very low level. This is precisely the case for the regulation of *recB* translation by Hfq, where a very abundant post-transcriptional regulator can effectively buffer the fluctuations of the low-abundant RecB enzyme. Taken together, these previous observations and our results could hint at a universal functional 'niche' for post-transcriptional regulators across a wide range of organisms.

## Materials and methods
### Strains and plasmids
All strains and plasmids used in the study are listed in *Supplementary file 1A, B* and are available upon request. The *E. coli* MG1655 and its derivatives were used for all conducted experiments except the one with arabinose-inducible expression where we used *E. coli* BW27783 background strain. The strains with RecB-HaloTag fusion were used in Halo-labelling experiments while the corresponding parental strains were used in smFISH experiments. MEK1329 and MEK1938 were built using the plasmid-mediated gene replacement method (*Link et al., 1997*) with pTOF24 derivative plasmids, pTOFΔ*recB* (*Darmon et al., 2010*) and pTOF*recB*-5′UTR, respectively. The deletion of *recB* and a 36-bp region (5′-TTAACGTGTTGAATCTGGACAGAAAATTAAGTTGAT-3′) was verified with Sanger sequencing. MEK1902 and MEK1457 were constructed with lambda red cloning technique using hfq_H1_P1 and hfq_H2_P2 primers (*Supplementary file 1C*). MEK1888 and MEK1449 were constructed with lambda red using ChiX_H1_P1 and ChiX_H2_P2 primers (*Supplementary file 1C*). *hfq* and *chiX* deletions were verified by PCR.

pIK02 was constructed to induce *recB* expression. The backbone was amplified with the primers oIK01/oIK02 from pBAD33 vector while the *recB* gene was amplified from *E. coli* chromosome using oIK03/oIK04 oligos (*Supplementary file 1C*). The PCR products were ligated using Gibson. Plasmid construction was verified by sequencing. pZA21-ChiX plasmid was constructed to allow overproduction of a small RNA ChiX. The plasmid, derived from the pZA21MCS (Expressys, Germany), was generated according to the protocol described here (*Iosub et al., 2020*) using chiX_ZA21 and pZA21_5P primers (*Supplementary file 1C*). To construct pTOF*recB*-5′UTR, the pTOF24 plasmid was digested with XhoI and SalI and ligated using Gibson assembly with the gBlock *recB*-5′UTR (*Supplementary file 1D*). The deletion of a 36-bp region was confirmed by sequencing.

### Growth media and conditions
In microscopy and population experiments, cell cultures were grown in M9-based medium supplemented with 0.2% glucose, 2 mM MgSO$_4$, 0.1 mM CaCl$_2$ and either 10% Luria Broth (*Figures 1 and 2*) or MEM amino acids (from Gibco) (*Figures 3–5*). In the *recB* induction smFISH experiments, 0.2%

glucose was replaced with 0.2% glycerol, and arabinose ranging from $10^{-5}$% to 1% was added to the medium. For mRNA lifetime measurements with smFISH, rifampicin (Sigma-Aldrich) was added to cell cultures to a final concentration of 500 µg/ml. In the DSB induction experiments, a sub-lethal concentration of ciprofloxacin 4 ng/ml was used. Ampicillin (50 µg/ml), kanamycin (50 µg/ml) or chloramphenicol (30 µg/ml) were added to cultures where appropriate.

Cell cultures were grown in medium overnight (14–16 hr) at 37°C. The overnight cultures were diluted (1:300) and grown at 37°C until the mid-exponential phase (optical density $OD_{600}$ = 0.2–0.3). Unless otherwise stated, cells were further treated according to the smFISH or Halo-labelling procedures described below. In the *recB* induction smFISH experiments, arabinose was added after 1 hr of overday growth, the cultures were grown for one more hour, and then samples were collected. For the smFISH experiments with rifampicin, the antibiotic was added once the cultures had reached $OD_{600}$ = 0.2. Cells were harvested at 1 min intervals and fixed in 3.2% formaldehyde. In the DSB induction experiments, ciprofloxacin was added at $OD_{600}$ ~ 0.1 and cell cultures continued growing for 1 hr. For mRNA quantification with smFISH, the cultures were grown for one more hour before being harvested and fixed in the final concentration of 3.2% formaldehyde. For protein quantification, the Halo-labelling protocol was followed as described below with ciprofloxacin being kept in the medium during labelling and washing steps.

## Single-molecule RNA FISH

smFISH experiments were carried out according to the established protocol (*Skinner et al., 2013*). Bacterial cells were grown as described above, fixed in formaldehyde solution, permeabilized in ethanol and hybridized with a *recB* specific set of TAMRA-labelled RNA FISH probes (LGC Biosearch Technologies). Probe sequences are listed in *Supplementary file 1F*. Unbound RNA FISH probes were removed with multiple washes, and then samples were visualized as described below.

## Halo-labelling

For single-molecule RecB protein quantification, we followed the Halo-labelling protocol described previously (*Lepore et al., 2019*). *E. coli* strains with RecB-HaloTag fusion, grown as described above, were labelled with Janelia Fluor 549 dye (purchased from Promega) for 1 hr, washed with aspiration pump (four to five times), fixed in 2.5% formaldehyde and mounted onto agar pads before imaging. In each experiment, a parental strain which does not have the HaloTag fusion (no HaloTag) was subjected to the protocol in parallel with the primary samples as a control.

## Microscopy set-up and conditions

Image acquisition was performed using an inverted fluorescence microscope (Nikon Ti-E) equipped with an EMCCD Camera (iXion Ultra 897, Andor), a Spectra X Line engine (Lumencor), dichronic mirror T590LPXR, 100× oil-immersion Plan Apo objective (NA 1.45, Nikon) and ×1.5 magnification lens (Nikon). A TRITC (ET545/30nm, Nikon) filter was used for imaging Halo-labelling experiments while smFISH data were acquired with an mCherry (ET572/35nm, Nikon) filter.

Once a protocol (smFISH or Halo-labelling) had been carried out, fixed cells were resuspended in 1X PBS and mounted onto a 2% agarose pad. The snapshots were acquired in bright-field and fluorescence channels. In all experiments, an electron-multiplying (EM) gain of 4 and an exposure time of 30 ms were used for bright-field imaging. In the fluorescence channel, smFISH data were acquired with a 2 s exposure time and a gain of 4, while RecB proteins were visualized with the same exposure time but EM gain 200. For each XY position on an agarose pad, a Z-stack of 6 images centred around the focal plane (total range of 1 µm with a step of 0.2 µm) was obtained in both channels. A set of multiple XY positions was acquired for each slide to visualize ~1000 cells per sample.

## Cell segmentation

Microscopy bright-field images were used to identify positions of bacterial cells with an automated MATLAB-based pipeline (*Jaramillo-Riveri et al., 2022*). Briefly, the segmentation algorithm is based on detection of cell edges by passing an image through a low-pass filter. In comparison to the original version, we applied this analysis to defocused bright-field images (instead of fluorescence signal from a constitutive reporter *Jaramillo-Riveri et al., 2022*) and tuned segmentation parameters for our conditions. The segmentation outputs were manually corrected by discarding misidentified cells

in the accompanied graphical interface. The corrected segmentation results were saved as MATLAB matrices (cell masks) containing mapping information between pixels and cell ID numbers. The cell masks were used as an input for the spot-finding software described below.

## Spot detection

The Spätzcell package was utilized to detect foci in fluorescent smFISH images (*Skinner et al., 2013*). In principle, the analysis consists of the following steps: (1) identification of local maxima above a chosen intensity threshold, (2) matching the maxima across frames in a Z-stack, and (3) performing 2D-Gaussian fitting of the detected maxima. Based on the peak height and spot intensity, computed from the fitting output, the specific signal was separated from false positive spots (*Figure 1—figure supplement 1a*). To identify the number of co-localized mRNAs, the integrated spot intensity profile was analysed as previously described (*Skinner et al., 2013*). Assuming that (1) probe hybridization is a probabilistic process, (2) binding each RNA FISH probe happens independently, and (3) in the majority of cases, due to low abundance, there is one mRNA per spot, it is expected that the integrated intensities of FISH probes bound to one mRNA are Gaussian distributed. In the case of two co-localized mRNAs, there are two independent binding processes and, therefore, a wider Gaussian distribution with twice higher mean and twice larger variance is expected. In fact, the integrated spot intensity profile had a main mode corresponding to a single mRNA per focus, and a second one representing a population of spots with two co-localized mRNAs (*Figure 1—figure supplement 1b*). Based on this model, the integrated spot intensity histograms were fitted to the sum of two Gaussian distributions (see *Equation 2*, where $a$, $b$, $c$, and $d$ are the fitting parameters), corresponding to one and two mRNA molecules per focus. An intensity equivalent corresponding to the integrated intensity of FISH probes in average bound to one mRNA was computed as a result of multiple-Gaussian fitting procedure (*Figure 1—figure supplement 1b*), and all identified spots were normalized by the one-mRNA equivalent.

$$\text{f}(x) = a\,e^{-\frac{(x-b)^2}{c^2}} + d\,e^{-\frac{(x-2b)^2}{2c^2}} \tag{2}$$

To detect fluorescent single-molecule foci in the Halo-labelling experiments, the first step of Spätzcell analysis was modified (similar to the modification for fluorescent protein detection implemented here *Wang et al., 2019*). For the Halo-labelling single-molecule data, we tested two modifications: (1) removing a Gaussian filter (that was used to smooth raw signal in the original version) or (2) calculating the Laplacian of a Gaussian-filtered image. Both versions gave similar and consistent quantification results. Peak height intensity profiles allowed the separation of specific signal from false positive spots (*Figure 2—figure supplement 1*). As a result of the low abundance of the protein of interest and single-molecule labelling, the analysis of integrated intensity was skipped for Halo-labelling images. All other steps of the Spätzcell software remained unchanged.

Total (mRNA or protein) molecule numbers per cell were obtained by matching spot positions to cell segmentation masks and quantifying the number of (mRNA or protein) spots within each cell. Molecule concentration was calculated for each cell as total number of (mRNA or protein) molecules (per cell) divided by the area of the cell ($1/\mu m^2$).

## Bacterial RNA extraction and RT-qPCR

Bacterial cells were grown as described earlier, harvested in the equivalent of *volume* × $OD_{600}$ = 2.5 ml and flash-frozen in liquid nitrogen. In the Halo-labelling experiments, samples for RNA extraction were grown at 37 °C and collected before the fixation step. Total RNA extraction was performed with the guanidium thiocyanate phenol protocol (*Iosub et al., 2021*). Primers for real-time qPCR experiments are listed in *Supplementary file 1E*. RT-qPCR quantification was carried out on 20 ng of RNA with the Luna One-Step RT-qPCR Kit (NEB). All qPCRs were performed in technical triplicates. The qPCR reactions were carried out using LightCycler 96 (Roche). The *rrfD* gene (5S rRNA) was used as a reference gene to compute a fold-change relative to a control sample with the $2^{-\Delta\Delta Ct}$ method. Outliers with the standard deviation larger than *std(Ct)* > 0.3 across technical replicates were removed from the analysis.

## Population growth and viability assays

Bacterial cells, seeded by inoculating from fresh colonies, were grown in an appropriate medium at 37°C overnight. The cultures were diluted (1:1000) and grown in a shaking incubator at 37°C. Optical density measurements were taken over the day at $OD_{600}$ with the interval of 15 min. The exponential phase of growth, $OD_{600}$ = [0.08, 0.4], was used for the fitting procedure. The fitting was performed with a linear regression model in MATLAB.

In the viability assays, cells were grown overnight in Luria Broth at 37°C in a shaking incubator. The optical density of the overnight cultures was normalized to $OD_{600}$ = 1.0. Tenfold serial dilutions were then performed in 96-well plates for the range of $OD_{600}$ from 1.0 to $10^{-7}$. 100 µl of cells of a specific dilution were plated onto LB agar plates containing ciprofloxacin and ampicillin at a concentration of 0–16 ng/ml and 100 µg/ml, respectively. For each condition, two dilutions were optimally chosen to have 30–300 colonies per plate. After 24 hr of incubation at 37°C, the colonies on each plate were counted and colony forming units (CFU) per ml were calculated using the following formula:

$$\frac{CFU}{ml} = \frac{\text{Number of colonies per plate x dilution}}{100\mu l} \qquad (3)$$

The survival factor was calculated by normalizing against the CFU/ml in the absence of ciprofloxacin.

## Data analysis of CRAC data

The CLASH data (NCBI Gene Expression Omnibus (GEO), accession number GSE123050) were analysed in the study (*Iosub et al., 2020*) with the entire pyCRAC pipeline available here (*Granneman, 2021a*; *Granneman, 2021b*). The SGR file, generated with CRAC_pipeline_PE.py, was used to plot the genome browser track in *Figure 4A*.

## Fitting and parameter estimation

The smFISH data from *Figure 1C* was conditioned on cell size (cell area <3.0 µm$^2$, *newborns*) and the updated distribution was fitted by a negative binomial distribution. In mathematical terms, a negative binomial *NB(r, p)* is defined by two parameters: *r*, the number of successful outcomes, and *p*, the probability of success. Mean and variance of *X* ~ *NB(r, p)* are given as:

$$\langle X \rangle = \frac{pr}{(1-p)} \qquad \sigma_X^2 = \frac{pr}{(1-p)^2} \qquad (4)$$

The probability density function of the fitted distribution was found with the maximum likelihood estimation method in MATLAB. The fitted parameters, *r* and *p*, are given as *r* = 0.335 and *p* = 0.513.

The average mRNA burst size *b* and transcription rate $k_m$ were calculated using the fitted parameters, *r* and *p*, and the *recB* mRNA degradation rate ($\gamma_m$ = 0.615 min$^{-1}$, measured in an independent time-course experiment: *Figure 1D*) according to the following analytical expressions (*Equations A4, A5* and *Equation 4*):

$$b = \frac{1-p}{p} = 0.95 \, molec \qquad (5)$$

$$k_m = \gamma_m r = 0.21 \, min^{-1} \qquad (6)$$

The RecB protein distribution from *Figure 2C* was conditioned on cell size (cell area <3.0 µm$^2$, *newborns*). Then, the average protein count number in *newborns* (~3.2 molecules per cell) was matched to the theoretical expression for the protein mean *Equation A4*. The rate of translation $k_p$ was calculated as follows (using protein removal rate, $\gamma_p$ = 0.015 min$^{-1}$, and mean mRNA number in *newborns*, $\langle m \rangle$ = 0.323):

$$k_p = \frac{\gamma_p \langle p \rangle}{\langle m \rangle} = 0.15 \, min^{-1} \qquad (7)$$

All parameters of the model are listed in *Supplementary file 1G*.

## Statistical analysis

Unless otherwise stated, histograms and RT-qPCR results represent the average across at least three replicated experiments and error bars reflect the maximum difference between the repeats. The Kullback–Leibler divergence ($D_{KL}$) between simulated and experimental distributions was calculated with the MATLAB function (*Razavi, 2025*). Comparison between the average protein or mRNA concentrations among different conditions was performed with a two-sample *t*-test and *P*-values were calculated in the MATLAB built-in function ttest2. Average translational efficiency was calculated as the ratio between the mean protein concentration $c_p$ and the mean mRNA concentration $c_m$ (*Figures 3F and 4F*). The error bars are standard deviations across replicated experiments.

## Stochastic simulations

Single trajectories of mRNA and protein abundance over time were generated for the reaction scheme *Equation A1* with the Gillespie algorithm, executed in MATLAB environment (*Abdennur, 2012*). Parameters used in simulations are listed in *Supplementary file 1G*. 10,000 simulations were performed for a given condition and probabilistic steady-state distributions were recovered for population snapshot data.

## Acknowledgements

We thank David Leach, Gerald Smith, Teppei Morita, and Hiroji Aiba for providing strains. We also thank Léna Le Quellec, Sebastian Jaramillo-Riveri, Alessia Lepore, Benura Azeroglu, James Holehouse, and Rachel Jackson for their technical support and assistance with cloning, image analysis, and modelling. We thank Livia Scorza from the Biological Research Data Management team for expert data curation. We are grateful to Ramon Grima, Rosalind Allen, David Leach, Peter Swain, and Johan Paulsson for fruitful discussions and generous advice.

## Additional information

### Funding

| Funder | Grant reference number | Author |
|---|---|---|
| Wellcome Trust | 10.35802/205008 | Meriem El Karoui |
| Biotechnology and Biological Sciences Research Council | BB/S008012/1 | Meriem El Karoui |
| Darwin Trust of Edinburgh | | Irina Kalita |
| Wellcome Trust | 10.35802/092076 | Sander Granneman |
| Medical Research Council | MR/R008205/1 | Sander Granneman |
| Wellcome Trust | 10.35802/102334 | Ira Alexandra Iosub |

The funders had no role in study design, data collection and interpretation, or the decision to submit the work for publication. For the purpose of Open Access, the authors have applied a CC BY public copyright license to any Author Accepted Manuscript version arising from this submission. Open access funding provided by Max Planck Society.

### Author contributions

Irina Kalita, Conceptualization, Software, Formal analysis, Investigation, Visualization, Methodology, Writing – original draft, Writing – review and editing; Ira Alexandra Iosub, Conceptualization, Formal analysis, Investigation, Visualization, Methodology, Writing – review and editing; Lorna McLaren, Louise Goossens, Investigation, Writing – review and editing; Sander Granneman, Conceptualization, Software, Formal analysis, Supervision, Funding acquisition, Methodology, Writing – review and editing; Meriem El Karoui, Conceptualization, Formal analysis, Supervision, Funding acquisition, Methodology, Writing – original draft, Writing – review and editing

## Author ORCIDs
Irina Kalita ![ORCID] https://orcid.org/0000-0002-0127-077X
Meriem El Karoui ![ORCID] https://orcid.org/0000-0003-2522-613X

Reviewer #1 (Public review): https://doi.org/10.7554/eLife.94918.3.sa1
Reviewer #2 (Public review): https://doi.org/10.7554/eLife.94918.3.sa2
Reviewer #3 (Public review): https://doi.org/10.7554/eLife.94918.3.sa3
Author response https://doi.org/10.7554/eLife.94918.3.sa4

## Additional files

### Supplementary files
MDAR checklist

Supplementary file 1. Supplementary tables and references.

### Data availability
All data that support this study including microscopy data, RT-qPCRs, growth rates, viability assays, and source codes for image analysis are available on Zenodo. The modified versions of the image analysis used for spot detection in Halo-labelling experiments and bright-field segmentation pipeline are also available on GitLab (SpotDetection_spatzcells, copy archived at *Kalita, 2025a*; CellSegmentation_MEK, copy archived at *Kalita, 2025b*). The CRAC/CLASH dataset (NCBI Gene Expression Omnibus ID GSE123050) (*Iosub et al., 2020*), used in this study, is available from GSE123050. The pipeline for the analysis of the CRAC/CLASH data is available from GitLab (sgrannem) and PyPI (g_ronimo). The article is additionally supported by *Supplementary file 1* containing Supplementary tables (A–G) with the lists of strains, plasmids, sequences of primers, gBlock, RNA FISH probes, and parameters of the model, and Supplementary references.

The following dataset was generated:

| Author(s) | Year | Dataset title | Dataset URL | Database and Identifier |
| --- | --- | --- | --- | --- |
| Kalita I, Iosub IA, McLaren L, Goossens L, Granneman S, Karoui EL | 2025 | An Hfq-dependent post-transcriptional mechanism fine tunes RecB expression in *Escherichia coli* | https://doi.org/10.5281/zenodo.15676159 | Zenodo, 10.5281/zenodo.15676159 |

The following previously published dataset was used:

| Author(s) | Year | Dataset title | Dataset URL | Database and Identifier |
| --- | --- | --- | --- | --- |
| Granneman S | 2020 | Hfq CLASH uncovers sRNA-target interaction networks linked to nutrient availability adaptation | https://www.ncbi.nlm.nih.gov/geo/query/acc.cgi?acc=GSE123050 | NCBI Gene Expression Omnibus, GSE123050 |

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

## Appendix 1

## A stochastic two-stage model of RecB expression

To model RecB expression, we applied a well-established stochastic mRNA–protein model of gene expression and inferred its parameters by fitting to the experimental data. Here, we briefly introduce the model, discuss its underlying assumptions and provide the analytical expressions used in the fitting procedure.

## Model description

A stochastic two-stage model of gene expression has been widely utilized to study fluctuations underlying protein production in prokaryotic and eukaryotic cells (*Shahrezaei and Swain, 2008*; *Thattai and van Oudenaarden, 2001*; *Paulsson, 2005*; *Friedman et al., 2006*; *Kumar et al., 2015*; *Smith and Grima, 2018*). Generally, the model is based on four biological processes: transcription of a gene by an RNA polymerase, translation of an mRNA by a ribosome, mRNA degradation and protein decay. As elsewhere, we assume a cell to be a well-stirred system of biochemical molecules, the interaction between which can be described with a common approach of chemical reactions (*van Kampen, 1992*). We focus on two species, mRNAs and proteins, and denote molecule numbers in a cell as $m$ and $p$, respectively.

Considering gene expression in bacterial cells allows several further assumptions to be made. First, transcription of bacterial genes was demonstrated to occur in bursts (*Golding et al., 2005*; *So et al., 2011*) and quite a few transcriptional models describing gene bursting have been proposed in the literature (*Paulsson, 2005*; *Perez-Carrasco et al., 2020*; *Klindziuk et al., 2020*; *Jones and Elf, 2018*). Thus, mRNA production is described as a zeroth-order chemical reaction where a gene is transcribed in mRNA bursts with a constant rate $k_m$ and the number of mRNAs per each burst $n$ is sampled from a geometric distribution with a mean size $b$ (*Peccoud and Ycart, 1995*; *Raj et al., 2006*; *Sanchez et al., 2013*). Secondly, based on a broadly accepted view of the exponential RNA degradation in bacteria (*Selinger et al., 2003*; *Mosteller et al., 1970*; *Schwartz et al., 1970*), mRNA decay is modelled as a first-order reaction with a constant rate $\gamma_m$. Furthermore, proteins in bacteria were shown to be diluted exclusively because of cell growth and division (*Taniguchi et al., 2010*; *Koch and Levy, 1955*). This process can be implicitly modelled by a first-order chemical reaction with a constant rate equal to growth rate $\gamma_p$. Finally, we assume that a protein is produced from an mRNA with a constant rate $k_p$. Thus, the entire reaction network for the two-stage model with burst transcription is summarized in the following schematic:

$$\emptyset \xrightarrow{k_m} nm \qquad\qquad m \xrightarrow{k_p} m + p$$
$$m \xrightarrow{\gamma_m} \emptyset \qquad\qquad p \xrightarrow{\gamma_p} \emptyset \tag{A1}$$

The dynamics of the mRNA and protein average levels (denoted as $\langle m \rangle$ and $\langle p \rangle$) is described by a set of the following rate equations:

$$\frac{d\langle m \rangle}{dt} = b k_m - \gamma_m \langle m \rangle \tag{A2}$$

$$\frac{d\langle p \rangle}{dt} = k_p \langle m \rangle - \gamma_p \langle p \rangle \tag{A3}$$

## Steady-state mRNA and protein mean and variance

The stationary solutions for mean mRNA and protein numbers can be found by imposing the steady-state conditions ($d\langle m \rangle/dt = 0$ and $d\langle p \rangle/dt = 0$) to the rate equations (*Equations A2 and A3*):

$$\langle m \rangle = b \frac{k_m}{\gamma_m} \qquad\qquad \langle p \rangle = b \frac{k_m k_p}{\gamma_m \gamma_p} \tag{A4}$$

As a result of the linearity of the system, the mRNA and protein variances can be found exactly by van Kampen's Ω-expansion (*van Kampen, 1992*; *Paulsson, 2004*; *Elf and Ehrenberg, 2003*).

$$\sigma_m^2 = (1+b)\langle m \rangle \qquad \sigma_p^2 = \langle p \rangle + \frac{k_p^2(1+b)\langle m \rangle}{\gamma_p(\gamma_m + \gamma_p)} \qquad \text{(A5)}$$

Finally, the mRNA and protein coefficients of variation (squared), defined as variance over squared mean, are given as follows (*Shahrezaei and Swain, 2008*; *Thattai and van Oudenaarden, 2001*; *Paulsson, 2005*; *Pedraza and Paulsson, 2008*):

$$CV_m^2 = \frac{\sigma_m^2}{\langle m \rangle^2} = \frac{1+b}{\langle m \rangle} \qquad \text{(A6)}$$

$$CV_p^2 = \frac{\sigma_p^2}{\langle p \rangle^2} = \frac{1}{\langle p \rangle} + \frac{1+b}{\langle m \rangle}\frac{\gamma_p}{\gamma_m + \gamma_p} \qquad \text{(A7)}$$

The analytical probability distribution of protein numbers under assumption of a long-lived protein relative to short-lived mRNA was derived here (*Shahrezaei and Swain, 2008*).

## Appendix 2

### Deterministic model of RecB expression under DNA damage conditions

In this section, we analytically describe the dynamics of RecB mRNA and protein levels after a perturbation made by DNA damage with a sub-lethal dose of ciprofloxacin. Taking into account different time-scales of the response in mRNA and protein levels, we address the following question: how long does the DNA damage need to be applied for to see changes (if any) in both species? We focus on the average mRNA and protein concentrations instead of molecule numbers because of cell elongation upon ciprofloxacin treatment (*Figure 3B, C*) and modify a two-stage model of RecB expression by including deterministic time dependency of cell volume, $V(t)$.

As discussed above, we consider the same set of reactions for two species of the system: mRNAs and proteins. While the processes of translation, active mRNA and protein degradation are described by first-order reactions with constant rates: $k_p$, $\gamma_m$, and $\gamma_{p(act)}$, respectively, mRNA transcription is given by a zeroth-order reaction. According to the law of mass action, we shall scale a zeroth-order reaction rate by the volume of the system, as $k_m V(t)$ (*van Kampen, 1992*; *Schnoerr et al., 2017*). We define the number of mRNA and protein molecules at time $t$ in a cell of volume $V(t)$ as $m(t)$ and $p(t)$, respectively. Then, the evolution of the average numbers of mRNA and protein molecules in time is described by the following rate equations:

$$\frac{d \langle m(t) \rangle}{dt} = k_m V(t) - \gamma_m \langle m(t) \rangle \tag{A8}$$

$$\frac{d \langle p(t) \rangle}{dt} = k_p \langle m(t) \rangle - \gamma_{p(act)} \langle p(t) \rangle \tag{A9}$$

We further define *recB* mRNA and RecB protein concentration as $c_m(t) = m(t)/V(t)$ and $c_p(t) = p(t)/V(t)$ and differentiate with respect to $t$:

$$\frac{dc_m(t)}{dt} = \frac{1}{V(t)} \frac{dm(t)}{dt} - \frac{c_m(t)}{V(t)} \frac{dV(t)}{dt} \tag{A10}$$

$$\frac{dc_p(t)}{dt} = \frac{1}{V(t)} \frac{dp(t)}{dt} - \frac{c_p(t)}{V(t)} \frac{dV(t)}{dt} \tag{A11}$$

In order to obtain time-dependent solutions for mRNA and protein concentrations, we divide the rate equations (*Equations A8 and A9*) by cell volume $V(t)$ and replace $dm(t)/dt$ and $dp(t)/dt$ with the expressions from the equations (*Equations A10 and A11*). Based on the experimental evidence (*Wang et al., 2010*; *Mir et al., 2011*), we assume the exponential dependence of cell volume from time, $V(t) = V_0 e^{\gamma t}$ , where $\gamma$ is a growth rate. Thus, we obtain the following ordinary differential equations for mRNA and protein concentrations:

$$\frac{d \langle c_m(t) \rangle}{dt} = k_m - (\gamma_m + \gamma) \langle c_m(t) \rangle \tag{A12}$$

$$\frac{d \langle c_p(t) \rangle}{dt} = k_p \langle c_m(t) \rangle - (\gamma_{p(act)} + \gamma) \langle c_p(t) \rangle \tag{A13}$$

By equating the left-hand sides of the equations to zero, we find the steady-state mRNA and protein concentrations, $c_m^{ss}$ and $c_p^{ss}$ :

$$c_m^{ss} = \frac{k_m}{\gamma_m + \gamma} \qquad c_p^{ss} = \frac{k_m}{(\gamma_m + \gamma)} \frac{k_p}{(\gamma_{p(act)} + \gamma)} \tag{A14}$$

Next, by assuming the initial conditions for mRNA and protein concentration to be $c_{m(t=0)}$ and $c_{p(t=0)}$, we obtain the time-dependent solutions for mRNA and protein concentrations:

$$\langle c_m(t) \rangle = \left( c_{m0} - \frac{k_m}{\gamma_m + \gamma} \right) e^{-(\gamma_m + \gamma)t} + \frac{k_m}{\gamma_m + \gamma} \tag{A15}$$

$$\langle c_p(t) \rangle = \left( c_{p0} + \frac{c_{m0}k_p}{\gamma_m - \gamma_{p(act)}} - \frac{k_p k_m}{(\gamma_m - \gamma_{p(act)})(\gamma_{p(act)} + \gamma)} \right) e^{-(\gamma_{p(act)} + \gamma)t} +$$
$$+ \left( -\frac{c_{m0}k_p}{\gamma_m - \gamma_{p(act)}} + \frac{k_p k_m}{(\gamma_m - \gamma_{p(act)})(\gamma_m + \gamma)} \right) e^{-(\gamma_m + \gamma)t} + \frac{k_p k_m}{(\gamma_{p(act)} + \gamma)(\gamma_m + \gamma)} \quad \text{(A16)}$$

The last expression can be simplified by taking into account that the RecB protein does not have an active degradation mechanism (**Figure 2D**). Thus, after imposing the condition of $\gamma_{p(act)} = 0$, we obtain a time-dependent solution for RecB protein concentration as follows:

$$\langle c_p(t) \rangle = \left( c_{p0} + \frac{c_{m0}k_p}{\gamma_m} - \frac{k_p k_m}{\gamma_m \gamma} \right) e^{-\gamma t} + \left( -\frac{c_{m0}k_p}{\gamma_m} + \frac{k_p k_m}{\gamma_m(\gamma_m + \gamma)} \right) e^{-(\gamma_m + \gamma)t} + \frac{k_p k_m}{\gamma(\gamma_m + \gamma)} \quad \text{(A17)}$$

From the expressions (**Equations A15 and A17**) we can now estimate characteristic time-scales needed for mRNA and protein levels to change to the new conditions. It is worth noting that although cells are elongated upon ciprofloxacin treatment, they do not slow down the growth (**Figure 3—figure supplement 1**). Thus, the analysis of the obtained equations shows that mRNA solution has one mode, $e^{-(\gamma_m + \gamma)t}$, and hence a characteristic time-scale can be estimated as $1/(\gamma_m + \gamma) \sim 1.6$ min (the results of growth rate measurements, $\gamma = \gamma_{cipro+} = 0.0165$ min$^{-1}$, are shown in **Figure 3—figure supplement 1c**). In contrast, the expression for protein concentration is given by two time-scales: $e^{-\gamma t}$ and $e^{-(\gamma_m + \gamma)t}$, with the slowest one $1/\gamma \sim 61$ min. This means that the perturbation needs to be applied for at least $\sim 1$ hr in order to be able to detect changes in protein concentrations. In our experiments, DSBs were induced for 2 hr to guarantee fulfilment of the time-scale requirement (**Figure 3**).

## Appendix 3

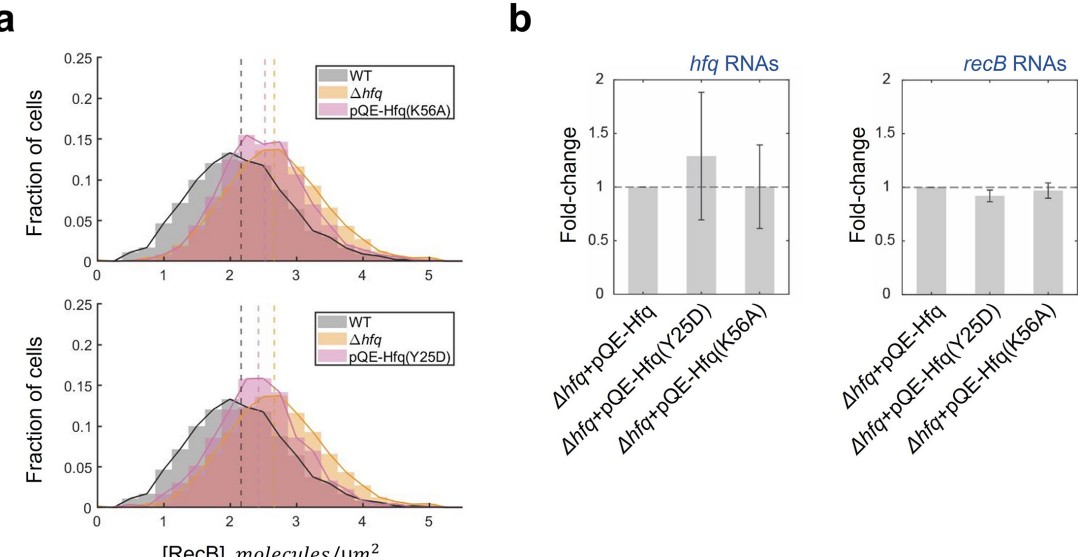

**Appendix 3—figure 1.** RecB quantification upon expression of Hfq with point mutations in proximal or distal binding faces. (**a**) RecB protein concentration distributions quantified in *hfq* mutants carrying a pQE80L-derivative plasmid with Hfq protein mutated either in proximal (K56A) or distal (Y25D) binding faces (*Zhang et al., 2013*). The histograms represent the average across two replicated experiments for each mutation. The RecB concentration histograms for wild type and *hfq* mutants are plotted as references in grey and orange, respectively. The dashed lines represent the mean RecB concentration in each condition. Significance was evaluated with two-sample *t*-test: *P*-values: 0.30(ns) for Δ*hfq* and Δ*hfq*+pQE-Hfq(K56A); 0.07(ns) for Δ*hfq* and Δhfq +pQE-Hfq(Y25D). (**b**) *hfq* and *recB* RNA levels quantified by RT-qPCR in Δ*hfq* cells carrying a pQE80L-derivative plasmid with a mutated Hfq protein. The results were normalized to the corresponding expression in Δ*hfq* cells carrying the backbone plasmid, pQE-Hfq.

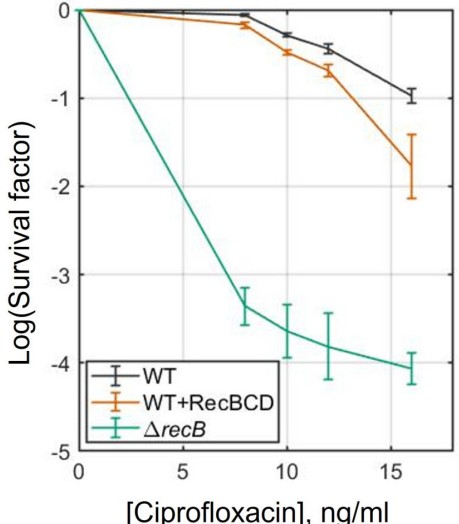

**Appendix 3—figure 2.** Toxicity of RecBCD over-expression upon DSB induction. Viability assays were performed for the wild type, wild type carrying RecBCD over-expression plasmid, pDWS2, and Δ*recB*. Cells were plated onto LB plates supplemented with ampicillin and either without or with 8/10/12/16 ng/ml of ciprofloxacin. The average survival factors were calculated for at least three replicated experiments while the error bars indicate standard estimation of the mean.

