## [Editor Report · eLife Assessment]

Combining experimental and computation approaches, this manuscript provides **convincing** evidence for a post-transcriptional mechanism that provides robust control over the protein expression level of RecB in *E. coli*. In addition to uncovering how DNA damage drives higher levels of RecB protein, this work also reveals **important** tenets for how broader mechanisms that suppress noise and underlie responsive tuning of protein levels can be achieved.

---

## [Referee Report · Reviewer #1 (Public review)]

Summary:

In this study the authors use an elegant set of single-molecule experiments to assess the transcriptional and post-transcriptional regulation of RecB. The question stems from a previous observation from the same lab, that RecB protein levels are low and not induced under DNA damage. The authors first show that recB transcript levels are low and have a short half-live. They further show that RecB levels are likely regulated via translational control. They provide evidence for low noise in RecB protein levels across cells and show that the translation of the mRNA increases under double-strand break conditions. Authors identify Hfq binding sites in the recbcd operon and show that Hfq regulates the levels of RecB protein without changing the mRNA levels. They suggest that RecB translation is directly controlled by Hfq binding to mRNA, as mutating one of the binding sites has a direct effect on RecB protein levels.

The implication of Hfq in regulation of RecB translation is important, and suggests mechanisms of cellular response to DNA damage that are beyond the canonically studied mechanisms (such as transcriptional regulation by LexA). Data are clearly presented and the writing is direct and easy to follow. Overall, the study is well-designed and provides novel insights into the regulation of RecB, that is part of the complex required to process break ends.

Comments on revisions:

All my comments are addressed - I congratulate the authors on this excellent work.

---

## [Referee Report · Reviewer #2 (Public review)]

Summary:

The authors carry out a careful and rigorous quantitative analysis of RecB transcript and protein levels at baseline and in response to DNA damage. Using single-molecule FISH and Halo-tagging in order to achieve sensitive measurements, they provide evidence that enhanced RecB protein levels in response to DNA damage are achieved through a post-transcriptional mechanism mediated by the La-like RNA binding protein, Hfq. In terms of biological relevance, the authors suggest that this mechanism provides a way to control the optimum level of RecB expression as both deletion and over-expression are deleterious. In addition, the proposed mechanism provides a new framework for understanding how transcriptional noise can be suppressed at the protein level.

Strengths:

Strengths of the manuscript include the rigorous approaches and orthogonal evidence to support the core conclusions, for example, the evidence that altering either Hfq or its recognition sequence on the RNA similarly enhance the protein to RNA ratio of RecB. The writing is clear and the experiments are well-controlled. The modeling approaches provide essential context to interpret the data, particularly given the small numbers of molecules per cell. The interpretations are careful and well supported. The findings

Weaknesses:

Future studies (and possibly new experimental tools) will be needed to provide further insight into the relevance of the findings to more subtle changes in RecB levels than that occurring in response to extensive DNA damage.

---

## [Referee Report · Reviewer #3 (Public review)]

Summary:

The work by Kalita et al. reports regulation of RecB expression by Hfq protein in *E. coli* cell. RecBCD is an essential complex for DNA repair and chromosome maintenance. The expression level needs to be regulated at low level under regular growth conditions but upregulated upon DNA damage. Through quantitative imaging, the authors demonstrate that recB mRNAs and proteins are expressed at low level under regular conditions. While the mRNA copy number demonstrates high noise level due to stochastic gene expression, the protein level is maintained at a lower noise level compared to expected value. Upon DNA damage, the authors claim that the recB mRNA concentration is decreased, however RecB protein level is compensated by higher translation efficiency. Through analyzing CLASH data on Hfq, they identified two Hfq binding sites on RecB polycistronic mRNA, one of which is localized at the ribosome binding site (RBS). Through measuring RecB mRNA and protein level in the ∆hfq cell, the authors conclude that binding of Hfq to the RBS region of recB mRNA suppresses translation of recB mRNA. This conclusion is further supported by the same measurement in the presence of Hfq sequestrator, the sRNA ChiX, and the deletion of the Hfq binding region on the mRNA.

Strengths:

(1) The manuscript is well-written and easy to understand.

(2) While there are reported cases of Hfq regulating translation of bound mRNAs, its effect on reducing translation noise is relatively new.

(3) The imaging and analysis are carefully performed with necessary controls.

Comments on revisions:

The authors have addressed my previous concerns.

---

## [Author Response]

The following is the authors’ response to the original reviews.

**Reviewer #2 (Public Review):**
The authors make a compelling case for the biological need to exquisitely control RecB levels, which they suggest is achieved by the pathway they have uncovered and described in this work. However, this conclusion is largely inferred as the authors only investigate the effect on cell survival in response to (high levels of) DNA damage and in response to two perturbations - genetic knock-out or over-expression, both of which are likely more dramatic than the range of expression levels observed in unstimulated and DNA damage conditions.

In the discussion of the updated version of the manuscript, we have clarified the limits of our interpretation of the role of the uncovered regulation.

Lines 411-417: “It is worth noting that the observed decrease in cell viability upon DNA damage was detected for relatively drastic perturbations such as *recB* deletion and RecBCD overexpression. Verifying these observations in the context of more subtle changes in RecB levels would be important for further investigation of the biological role of the uncovered regulation mechanism. However, the extremely low numbers of RecB proteins make altering its abundance in a refined, controlled, and homogeneous across cells manner extremely challenging and would require the development of novel synthetic biology tools.”

**Reviewer #3 (Public Review):**
The major weaknesses include a lack of mechanistic depth, and part of the conclusions are not fully supported by the data.(1) Mechanistically, it is still unclear why upon DNA damage, translation level of recB mRNA increases, which makes the story less complete. The authors mention in the Discussion that a moderate (30%) decrease in Hfq protein was observed in previous study, which may explain the loss of translation repression on recB. However, given that this mRNA exists in very low copy number (a few per cell) and that Hfq copy number is on the order of a few hundred to a few thousand, it's unclear how 30% decrease in the protein level should resides a significant change in its regulation of recB mRNA.

We agree that the entire mechanistic pathway controlling *recB* expression may be not limited to just Hfq involvement. We have performed additional experiments, proposed by the reviewer, suggesting that a small RNA might be involved (see below, response to comments 3&4). However, we consider that the full characterisation of all players is beyond the scope of this manuscript. In addition to describing the new data (see below), we expanded the discussion to explain more precisely why changes in Hfq abundance upon DNA damage may impact RecB translation.

Lines 384-391: “A modest decrease (~30%) in Hfq protein abundance has been seen in a proteomic study in *E. coli* upon DSB induction with ciprofloxacin (DOI: 10.1016/j.jprot.2018.03.002). While Hfq is a highly abundant protein, it has many mRNA and sRNA targets, some of which are also present in large amounts (DOI: 10.1046/j.1365-2958.2003.03734.x). As recently shown, the competition among the targets over Hfq proteins results in unequal (across various targets) outcomes, where the targets with higher Hfq binding affinity have an advantage over the ones with less efficient binding (DOI: 10.1016/j.celrep.2020.02.016). In line with these findings, it is conceivable that even modest changes in Hfq availability could result in significant changes in gene expression, and this could explain the increased translational efficiency of RecB under DNA damage conditions. “

*(2) Based on the experiment and the model, Hfq regulates translation of recB gene through binding to the RBS of the upstream ptrA gene through translation coupling. In this case, one would expect that the behavior of ptrA gene expression and its response to Hfq regulation would be quite similar to recB. Performing the same measurement on ptrA gene expression in the presence and absence of Hfq would strengthen the conclusion and model.*

Indeed, based on our model, we expect PtrA expression to be regulated by Hfq in a similar manner to RecB. However, the product encoded by the *ptrA* gene, Protease III, (i) has been poorly characterised; (ii) unlike RecB, is located in the periplasm (DOI: 10.1128/jb.149.3.1027-1033.1982); and (iii) is not involved in any DNA repair pathway. Therefore, analysing PtrA expression would take us away from the key questions of our study.

(3) The authors agree that they cannot exclude the possibility of sRNA being involved in the translation regulation. However, this can be tested by performing the imaging experiments in the presence of Hfq proximal face mutations, which largely disrupt binding of sRNAs.(4) The data on construct with a long region of Hfq binding site on recB mRNA deleted is less convincing. There is no control to show that removing this sequence region itself has no effect on translation, and the effect is solely due to the lack of Hfq binding. A better experiment would be using a Hfq distal face mutant that is deficient in binding to the ARN motifs.

We performed the requested experiments. We included this data in the manuscript in the supplementary figure (Figure S11), and our interpretation in the discussion.

Lines 354-378: “While a few recent studies have shown evidence for direct gene regulation by Hfq in a sRNA-independent manner (DOI: 10.1101/gad.302547.117; DOI: 10.1111/mmi.14799; DOI: 10.1371/journal.pgen.1004440; DOI: 10.1111/mmi.12961; DOI: 10.1038/emboj.2013.205), we attempted to investigate whether a small RNA could be involved in the Hfq-mediated regulation of RecB expression. We tested Hfq mutants containing point mutations in the proximal and distal sides of the protein, which were shown to disrupt either binding with sRNAs or with ARN motifs of mRNA targets, respectively [DOI: 10.1016/j.jmb.2013.01.006, DOI: 10.3389/fcimb.2023.1282258]. Hfq mutated in either proximal (K56A) or distal (Y25D) faces were expressed from a plasmid in a ∆*hfq* background. In both cases, Hfq expression was confirmed with qPCR and did not affect *recB* mRNA levels (Supplementary Figure S11b). When the proximal Hfq binding side (K56A) was disrupted, RecB protein concentration was nearly similar to that obtained in a ∆*hfq* mutant (Supplementary Figure S11a, top panel). This observation suggests that the repression of RecB translation requires the proximal side of Hfq, and that a small RNA is likely to be involved as small RNAs (Class I and Class II) were shown to predominantly interact with the proximal face of Hfq [DOI: 10.15252/embj.201591569]. When we expressed Hfq mutated in the distal face (Y25D) which is deficient in binding to mRNAs, less efficient repression of RecB translation was detected (Supplementary Figure S11a, bottom panel). This suggests that RecB mRNA interacts with Hfq at this position. We did not observe full de-repression to the ∆*hfq* level, which might be explained by residual capacity of Hfq to bind its *recB* mRNA target in the point mutant (Y25D) (either via the distal face with less affinity or via the lateral rim Hfq interface).”

Taken together, these results suggest that Hfq binds to *recB* mRNA and that a small RNA might contribute to the regulation although this sRNA has not been identified.

(5) Ln 249-251: The authors claim that the stability of recB mRNA is not changed in ∆hfq simply based on the steady-state mRNA level. To claim so, the lifetime needs to be measured in the absence of Hfq.

We measured *recB* lifetime in the absence of Hfq in a time-course experiment where transcription initiation was inhibited with rifampicin and mRNA abundance was quantified with RT-qPCR. The results confirmed that *recB* mRNA lifetime in *hfq* mutants is similar to the one in the wild type (Figure S7d, referred to the line 263 of the manuscript).

(6) What's the labeling efficiency of Halo-tag? If not 100% labeled, is it considered in the protein number quantification? Is the protein copy number quantification through imaging calibrated by an independent method? Does Halo tag affect the protein translation or degradation?

Our previous study (DOI: 10.1038/s41598-019-44278-0) described a detailed characterization of the HaloTag labelling technique for quantifying low-copy proteins in single *E. coli* cells using RecB as a test case.

In that study, we showed complete quantitative agreement of RecB quantification between two fully independent methods: HaloTag-based labelling with cell fixation and RecB-sfGFP combined with a microfluidic device that lowers protein diffusion in the bacterial cytoplasm. This second method had previously been validated for protein quantification (DOI: 10.1038/ncomms11641) and provides detection of 80-90% of the labelled protein. Additionally, in our protocol, immediate chemical fixation of cells after the labelling and quick washing steps ensure that new, unlabelled RecB proteins are not produced. We, therefore, conclude that our approach to RecB detection is highly reliable and sufficient for comparing RecB production in different conditions and mutants.

The RecB-HaloTag construct has been designed for minimal impact on RecB production and function. The HaloTag is translationally fused to RecB in a loop positioned after the serine present at position 47 where it is unlikely to interfere with (i) the formation of RecBCD complex (based on RecBCD structure, DOI: 10.1038/nature02988), (ii) the initiation of translation (as it is far away from the 5’UTR and the beginning of the open reading frame) and (iii) conventional C-terminalassociated mechanisms of protein degradation (DOI: 10.15252/msb.20199208). In our manuscript, we showed that the RecB-HaloTag degradation rate is similar to the dilution rate due to bacterial growth. This is in line with a recent study on unlabelled proteins, which shows that RecB’s lifetime is set by the cellular growth rate (DOI: 10.1101/2022.08.01.502339).

Furthermore, we have demonstrated (DOI: 10.1038/s41598-019-44278-0) that (i) bacterial growth is not affected by replacing the native RecB with RecB-HaloTag, (ii) RecB-HaloTag is fully functional upon DNA damage, and (iii) no proteolytic processing of the RecB-HaloTag is detected by Western blot.

These results suggest that RecB expression and functionality are unlikely to be affected by the translational HaloTag insertion at Ser-47 in RecB.

In the revised version of the manuscript, we have added information about the construct and discuss the reliability of the quantification.

Lines 141-152: “To determine whether the mRNA fluctuations we observed are transmitted to the protein level, we quantified RecB protein abundance with singlemolecule accuracy in fixed individual cells using the Halo self-labelling tag (Fig. 2A&B).

The HaloTag is translationally fused to RecB in a loop after Ser47(DOI: 10.1038/s41598-019-44278-0) where it is unlikely to interfere with the formation of RecBCD complex (DOI: 10.1038/nature02988), the initiation of translation and conventional C-terminal-associated mechanisms of protein degradation (DOI: 10.15252/msb.20199208). Consistent with minimal impact on RecB production and function, bacterial growth was not affected by replacing the native RecB with RecBHaloTag, the fusion was fully functional upon DNA damage and no proteolytic processing of the construct was detected (DOI: 10.1038/s41598-019-44278-0). To ensure reliable quantification in bacteria with HaloTag labelling, the technique was previously verified with an independent imaging method and resulted in > 80% labelling efficiency (DOI: 10.1038/s41598-019-44278-0, DOI: 10.1038/ncomms11641)*.* In order to minimize the number of newly produced unlabelled RecB proteins, labelling and quick washing steps were followed by immediate chemical fixation of cells.”

Lines 164-168: “Comparison to the population growth rate [in these conditions (0.017 1/min)] suggests that RecB protein is stable and effectively removed only as a result of dilution and molecule partitioning between daughter cells. This result is consistent with a recent high-throughput study on protein turnover rates in *E. coli,* where the lifetime of RecB proteins was shown to be set by the doubling time (DOI: 10.1038/s41467-024-49920-8).”

(7) Upper panel of Fig S8a is redundant as in Fig 5B. Seems that Fig S8d is not described in the text.

We have now stated in the legend of Fig S8a that the data in the upper panel were taken from Fig 5B to visually facilitate the comparison with the results given in the lower panel. We also noticed that we did not specify that in the upper panel in Fig S9a (the data in the upper panel of Fig S9a was taken from Fig 5C for the same reason). We added this clarification to the legend of the Fig S9 as well.

We referred to the Fig S8d in the main text.

Lines 283-284: “We confirmed the functionality of the Hfq protein expressed from the pQE-Hfq plasmid in our experimental conditions (Fig. S8d).”

**Reviewer #1 (Recommendations For The Authors):**
(1) Experimental regime to measure protein and mRNA levels.(a) Authors expose cells to ciprofloxacin for 2 hrs. They provide a justification via a mathematical model. However, in the absence of a measurement of protein and mRNA across time, it is unclear whether this single time point is sufficient to make the conclusion on RecB induction under double-strand break.

In our experiments, we only aimed to compare *recB* mRNA and RecB protein levels in two steady-state conditions: no DNA damage and DNA damage caused by sublethal levels of ciprofloxacin. We did not aim to look at RecB dynamic regulation from nondamaged to damaged conditions – this would indeed require additional measurements at different time points. We revised this part of the results to ensure that our conclusions are stated as steady-state measurements and not as dynamic changes.

Line 203-205: “We used mathematical modelling to verify that two hours of antibiotic exposure was sufficient to detect changes in mRNA and protein levels and for RecB mRNA and protein levels to reach a new steady state in the presence of DNA damage.”

(b) Authors use cell area to account for the elongation under damage conditions. However, it is unclear whether the number of copies of the recB gene are similar across these elongated cells. Hence, authors should report mRNA and protein levels with respect to the number of gene copies of RecB or chromosome number as well.

Based on the experiments in DNA damaging conditions, our main conclusion is that the average translational efficiency of RecB is increased in perturbed conditions. We believe that this conclusion is well supported by our measurements and that it does not require information about the copy number of the *recB* gene but only the concentration of mRNA and protein. We did observe lower *recB* mRNA concentration upon DNA damage in comparison to the untreated conditions, which may be due to a lower concentration of genomic DNA in elongated cells upon DNA damage, as we mention in lines (221-223).

Our calculation of translation efficiency could be affected by variations of mRNA concentration across cells in the dataset. For example, longer cells that are potentially more affected by DNA damage could have lower concentrations of mRNA. We verified that this is not the case, as *recB* mRNA concentration is constant across cell size distribution (see the figure below or Figure S5a from Supplementary Information).

Therefore, we do not think that the measurements of *recB* gene copy would change our conclusions. We agree that measuring *recB* gene copies could help to investigate the reason behind the lower *recB* mRNA concentration under the perturbed conditions as this could be due to lower DNA content or due to shortage of resources (such as RNA polymerases). However, this is a side observation we made rather than a critical result, whose investigation is beyond the scope of this manuscript.

**Author response image 1. sa4fig1:** 

(2) RecB as a proxy for RecBCD. Authors suggest that RecB levels are regulated by hfq. However, how does this regulatory circuit affect the levels of RecC and RecD? Ratio of the three proteins has been shown to be important for the function of the complex.

A full discussion of RecBCD complex formation regulation would require a complete quantitative model based on precise information on the dynamic of the complex formation, which is currently lacking.

We can however offer the following (speculative) suggestions assuming that all three subunits are present in similar abundance in native conditions (DOI: 10.1038/s41598019-44278-0 for RecB and RecC). As the complex is formed in 1:1:1 ratio (DOI: 10.1038/nature02988), we propose that the regulation mechanism of RecB expression affects complex formation in the following way. If the RecB abundance becomes lower than the level of RecC and RecD subunits, the complex formation would be limited by the number of available RecB subunits and hence the number of functional RecBCDs will be decreased. On the contrary, if the number of RecB is higher than the baseline, then, especially in the context of low numbers, we would expect that the probability of forming a complex RecBC (and then RecBCD) will be increased. Based on this simple explanation, we might speculate that regulation of RecB expression may be sufficient to regulate RecB levels and RecBCD complex formation. However, we feel that this argument is too speculative to be added to the manuscript.

(3) Role of Hfq in RecB regulation. While authors show the role of hfq in recB translation regulation in non-damage conditions, it is unclear as to how this regulation occurs under damage conditions.(a) Have the author carried out recB mRNA and protein measurement in hfqdeleted cells under ciprofloxacin treatment?

We attempted to perform experiments in *hfq* mutants under ciprofloxacin treatment. However, the cells exhibited a very strong and pleiotropic phenotype: they had large size variability and shape changes and were also frequently lysing. Therefore, we did not proceed with mRNA and protein quantification because the data would not have been reliable.

(b) How do the authors propose that Hfq regulation is alleviated under conditions of DNA damage, when RecB translation efficiency increases?

We propose that Hfq could be involved in a more global response to DNA damage as follows.

Based on a proteomic study where Hfq protein abundance has been found to decrease (~ 30%) upon DSB induction with ciprofloxacin (DOI: 10.1016/j.jprot.2018.03.002), we suggest that this could explain the increased translational efficiency of RecB. While Hfq is a highly abundant protein, it has many targets (mRNA and sRNA), some of which are also highly abundant. Therefore the competition among the targets over Hfq proteins results in unequal (across various targets) outcomes (DOI: 10.1046/j.13652958.2003.03734.x), where the targets with higher Hfq binding affinity have an advantage over the ones with less efficient binding. We reason that upon DNA damage, a moderate decrease in the Hfq protein abundance (30%) can lead to a similar competition among Hfq targets where high-affinity targets outcompete low-affinity ones as well as low-abundant ones (such as *recB* mRNAs). Thus, the regulation of lowabundant targets of Hfq by moderate perturbations of Hfq protein level is a potential explanation for the change in RecB translation that we have observed. Potential reasons behind the changes of Hfq levels upon DNA damage would be interesting to explore, however this would require a completely different approach and is beyond the scope of this manuscript.

We have modified the text of the discussion to explain our reasoning:

Lines 384-391: “A modest decrease (~30%) in Hfq protein abundance has been seen in a proteomic study in *E. coli* upon DSB induction with ciprofloxacin (DOI: 10.1016/j.jprot.2018.03.002). While Hfq is a highly abundant protein, it has many mRNA and sRNA targets, some of which are also present in large amounts (DOI: 10.1046/j.1365-2958.2003.03734.x). As recently shown, the competition among the targets over Hfq proteins results in unequal (across various targets) outcomes, where the targets with higher Hfq binding affinity have an advantage over the ones with less efficient binding (DOI: 10.1016/j.celrep.2020.02.016). In line with these findings, it is conceivable that even modest changes in Hfq availability could result in significant changes in gene expression, and this could explain the increased translational efficiency of RecB under DNA damage conditions.”

(c) Is there any growth phenotype associated with recB mutant where hfq binding is disrupted in damage and non-damage conditions? Does this mutation affect cell viability when over-expressed or under conditions of ciprofloxacin exposure?

We checked the phenotype and did not detect any difference in growth or cell viability affecting the *recB*-5 UTR* mutants either in normal conditions or upon exposure to ciprofloxacin. However, this is expected because the repair capacity is associated with RecB protein abundance and in this mutant, while translational efficiency of *recB* mRNA increases, the level of RecB proteins remains similar to the wild-type (Figure 5E).

Minor points:(1) Introduction - authors should also discuss the role of RecFOR at sites of fork stalling, a likely predominant pathway for break generated at such sites.

The manuscript focuses on the repair of DNA double-strand breaks (DSBs). RecFOR plays a very important role in the repair of stalled forks because of single-strand gaps but is not involved in the repair of DSBs (DOI: 10.1038/35003501). We have modified the beginning of the introduction to mention the role of RecFOR.

Lines 35-39: “For instance, replication forks often encounter obstacles leading to fork reversal, accumulation of gaps that are repaired by the RecFOR pathway (DOI: 10.1038/35003501) or breakage which has been shown to result in spontaneous DSBs in 18% of wild-type *Escherichia coli* cells in each generation (DOI: 10.1371/journal.pgen.1007256), underscoring the crucial need to repair these breaks to ensure faithful DNA replication.”

(2) Methods: The authors refer to previous papers for the method used for single RNA molecule detection. More information needs to be provided in the present manuscript to explain how single molecule detection was achieved.

We added additional information in the method section on the fitting procedure allowing quantifying the number of mRNAs per detected focus.

Lines 515-530: “Based on the peak height and spot intensity, computed from the fitting output, the specific signal was separated from false positive spots (Fig. S1a). To identify the number of co-localized mRNAs, the integrated spot intensity profile was analyzed as previously described (DOI: 10.1038/nprot.2013.066). Assuming that (i) probe hybridization is a probabilistic process, (ii) binding each RNA FISH probe happens independently, and (iii) in the majority of cases, due to low-abundance, there is one mRNA per spot, it is expected that the integrated intensities of FISH probes bound to one mRNA are Gaussian distributed. In the case of two co-localized mRNAs, there are two independent binding processes and, therefore, a wider Gaussian distribution with twice higher mean and twice larger variance is expected. In fact, the integrated spot intensity profile had a main mode corresponding to a single mRNA per focus, and a second one representing a population of spots with two co-localized mRNAs (Fig. S1b). Based on this model, the integrated spot intensity histograms were fitted to the sum of two Gaussian distributions (see equation below where *a*, *b*, *c*, and *d* are the fitting parameters), corresponding to one and two mRNA molecules per focus. An intensity equivalent corresponding to the integrated intensity of FISH probes in average bound to one mRNA was computed as a result of multiple-Gaussian fitting procedure (Fig. S1b), and all identified spots were normalized by the one-mRNA equivalent.

\begin{document}$f(x)=a e^{-\frac{(x-b)^{2}}{c^{2}}}+d e^{-\frac{(x-2 b)^{2}}{2^{2}}}$\end{document}

**Reviewer #2 (Recommendations For The Authors):**
Overall the work is carefully executed and highly compelling, providing strong support for the conclusions put forth by the authors.One point: the potential biological consequences of the post-transcriptional mechanism uncovered in the work would be enhanced if the authors could (1) tune RecB protein levels and (2) directly monitor the role that RecB plays in generating single-standed DNA at DSBs.

We agree that testing viability of cells in case of tunable changes in RecB levels would be important to further investigate the biological role of the uncovered regulation mechanism. However, this is a very challenging experiment as it is technically difficult to alter the low number of RecB proteins in a controlled and homogeneous across-cell manner, and it would require the development of precisely tunable and very lowabundant synthetic designs.

We did monitor real-time RecB dynamics by tracking single molecules in live *E. coli* cells in a different study (DOI: 10.1101/2023.12.22.573010) that is currently under revision. There, reduced motility of RecB proteins was observed upon DSB induction indicating that RecB is recruited to DNA to start the repair process.